# HESS Opinion: Floods and droughts - Are land use, soil management, and landscape hydrology more significant drivers than increasing CO₂?

Karl Auerswald[1], Juergen Geist[1], John N. Quinton[2], Peter Fiener[3]

[1]School of Life Sciences, Technical University of Munich, Freising, 85354, Germany
[2]Lancaster Environment Centre, Lancaster University, Lancaster, LA1 4YQ, UK
[3]Working Group Water and Soil Resource Research, University of Augsburg, Augsburg, 86159, Germany

*Correspondence to*: John Quinton (j.quinton@lancaster.ac.uk)

**Abstract.** Floods, droughts, and heatwaves are increasing globally. This is typically attributed to CO₂-driven climate change.
However, at the global scale, CO₂-driven climate change neither reduces precipitation nor adequately explains droughts despite the modest increase in evapotranspiration due to temperature rise. Past land-use changes, particularly soil sealing, compaction, and drainage, are likely more significant for water losses by runoff leading to flooding and water scarcity. The importance of these processes is generally poorly addressed in modeling because hydrological models rarely reflect lateral fluxes in the atmosphere, on the soil surface, and in the soil. Land use is only considered in coarse categories, and neighborhood effects and
feedback mechanisms are neglected. However, even if models fail and we cannot create landscape experiments, there is sufficient evidence that land use is a key driver of the problem and of the solution to mitigate floods, droughts, and heatwaves. Addressing land-use changes is imperative as they persist even with zero net CO₂ emissions, making the world more vulnerable.

## 1 Introduction

Reports of severe storms, catastrophic floods like the Simbach event in southern Germany (Brandhuber et al., 2017; Mayr et al., 2020), and tragic events such as the Ahrtal floods in western Germany, which caused over 150 casualties (Mohr et al., 2023), are increasingly common. These occurrences, alongside water shortages, droughts, and heatwaves (Ciais et al., 2005; Miralles et al., 2019), suggest a significant alteration in landscape hydrology, often attributed to CO₂-driven climate change. In public discourse, explanations for these climate-driven changes often revolve around statements such as "The soil dries out
because of the heat" or "...because air humidity is so low," as seen in the German Drought Monitor (https://www.ufz.de/index.php?en=37937, Boing et al., 2022). While these statements correlate with observations, they offer only circular reasoning, lacking a causal explanation. This can easily be recognized because the logic of the sentences can be reversed and still holds: "It is so hot because the soil is dry" (García-García et al., 2024). Even the plausible sentence "the soil is dry because it hasn't rained for a long time" is at least partly circular reasoning since terrestrial evapotranspiration globally

contributes 40 % to terrestrial precipitation, with 57 % of the terrestrial evapotranspiration being recycled (van der Ent et al., 2010). Recycling ratios are higher in summer when oceans are cold relative to the land and in areas of low precipitation (van der Ent and Tuinenburg, 2017). Moisture recycling becomes especially large where large, intact woodland exists, extending recycling over thousands of square kilometers (Makarieva et al., 2013a; 2014). These values of moisture recycling, although large, are strongly biased towards minimum values because they only consider falling precipitation (snow, hail, and rain) while

they neglect occult precipitation (dew, fog, rime), which often is of local, recycled origin (Kaseke et al., 2017). Occult precipitation can be significant and reach several hundred millimeters per year (Zimmermann and Zimmermann, 2002; Ingraham and Mark, 2000; Migała et al., 2002; Jacobs et al., 2006).

To discern between spurious and causal relations, it is crucial to consider that the water and energy balances are interconnected, sharing evapotranspiration as a common variable (Allen et al., 1998).

The water balance equation in its simplest version is:

$$P - ET - Q - \Delta S = 0,  \tag{1}$$

with $P$ indicating precipitation, $ET$ evapotranspiration or latent heat, when considered in the energy balance, $Q$ runoff either laterally (surface or subsurface runoff) or vertically (groundwater recharge), and $\Delta S$ is the variable filling of the soil store. The equation applies to all scales even though the relative importance of the different terms changes with temporal and spatial

scale.

The simplest version of the energy balance without lateral energy fluxes is expressed as:

$$R_s \times (1 - \alpha) + R_{nl} - \lambda \times ET - G - H = 0.  \tag{2}$$

The variables indicate incipient short-wave radiation ($R_s$), albedo ($\alpha$), which is the fraction of reflected short-wave radiation, the net effect of incoming and outgoing long-wave radiation ($R_{nl}$), the enthalpy of evaporation ($\lambda$), the sensible heat flux ($H$),

and the soil heat flux ($G$).

$CO_2$-driven climate change directly impacts only $R_{nl}$ in equation 2, while land use intentionally influences $ET$, $Q$, $\Delta S$, $\alpha$, and $G$, affecting both equation 1 and 2 if we ignore the many cascading effects that follow in both cases. Albedo, for instance, is about 30 % larger for a straw-covered than for a bare soil surface (Sharrett and Campbell, 1994). A straw cover would allow every farmer to preserve soil moisture for crops because less energy from short-wave radiation would be available to drive

evapotranspiration. For France, it was estimated that, during the centennial European heat wave in August 2003, if the farmers had left the straw from grain harvest on the soil rather than tilling it in, the change of albedo would have lowed temperature country-wide on average by 2 K (Davin et al., 2014). This heatwave was the deadliest natural disaster in Europe in the last few centuries, with the death toll exceeding 70,000 in Europe and about 20,000 in France alone (Robine et al., 2008). Further direct effects of a straw cover would have contributed to a shorter, less intense drought due to: lower soil humidity efflux; lower

capillary rise to the evaporating surface due to the physical barrier by the straw cover; better infiltration during heavy rain due to less surface crusting; less erosion; and more dew formation due to better thermal isolation reducing the soil heat flux during

the night. Given the strong influence of soil and soil use over the water and energy balances, there is potential to compensate for some of the adverse effects that $CO_2$ increase has on terrestrial environments. However, this option is not commonly used. Instead, land use is another principal driver of floods, droughts, and heatwaves.

In this paper, we demonstrate and compare the $CO_2$-driven and land-use-driven climate change on floods and droughts to highlight the significant, but rarely considered, effects of land management on landscape hydrology, including droughts and floods. Floods encompass flashfloods and fluvial floods. We will exemplify this for Bavaria (southern Germany) for two reasons. (i) We have access to a large consistent monitoring and modelling data set regarding climate, land use and management and hydrology, and (ii) the region represents a typical Mid-European setting with predominantly agricultural land use (cropland

and livestock farming) while also featuring extensive forest, urban areas, and protected natural reserves. The general effects we show for southern Germany will occur globally in most regions, with obvious regional differences in land use setting and agricultural management, for example, by differences in agricultural machinery weights, the soil sealing area, or the road network's density. A short description of the example area can be found in the supplement.

## 2 $CO_2$-driven climate change

Globally, an increase in annual precipitation by 2 to 3 % per kelvin temperature increase can be expected (Lambert and Webb, 2008; Roderick et al., 2014; Bürger et al., 2014; Skliris et al., 2016). This value results from a 7 % $K^{-1}$ increase in the moisture-carrying capacity of the air, called the Clausius-Clapeyron (CC) rate, and constant energy provided by radiation driving evapotranspiration. Individual rains require a saturated atmosphere follow the CC rate and during intense rains, the CC rate can even be exceeded. The increase in rainfall intensity can reach 14 % $K^{-1}$, referred to as the super-CC rate (Westra et al.,

2014). This effect is particularly pronounced at temperatures ranging from 12 °C to 22 °C (Lenderink and van Meijgaard, 2008). The increase in rainfall intensity exceeding the CC rate can be attributed to an intensification of cloud dynamics (Loriaux et al., 2013; Westra et al., 2014), sometimes coupled with the transition from stratiform to convective regimes (Haerter and Berg, 2009). This increase in rain intensity arises from higher surface temperatures and, in particular, from the higher release of sensible heat during condensation of moister air. This additional energy augments the updraft strength of the

convective cell (Loriaux et al., 2013; Westra et al., 2014), leading to an increased influx of moist air, which is further enhanced due to the loss of vapor volume due to rain condensation (Makarieva et al., 2013b).

In addition to the general trends of rising temperature and increasing precipitation, many studies suggest that the unusually persistent and amplified disturbances in the jet stream are associated with persistent extreme weather events leading to floods or droughts. These persistent events have been related to high-amplitude quasi-stationary atmospheric Rossby waves resulting

from quasi-resonant amplification. However, there is considerable variation among climate models regarding this effect. Some predict a near-tripling of quasi-resonant amplification events by the end of the century, while others predict a potential decrease (Mann et al., 2018).

In Bavaria, the changes for air temperature during the last seven decades have been pronounced as annual mean temperature has risen from about 7.5°C to 9°C. Temperatures started to change circa 1980 and increased almost linearly (Fig. 1). Spatially, this increase was uniform. The temporal changes in precipitation, evapotranspiration, direct runoff, and groundwater recharge were minor compared to the spatial variation and without a clear trend (Fig. 1). Between 1950 and 2015, modeled groundwater recharge decreased by a total of 22 mm yr$^{-1}$ (i.e., on average, it decreased by 0.33 mm yr$^{-1}$ every year during the last 65 years), and actual evapotranspiration decreased by a total of 7 mm yr$^{-1}$, while precipitation decreased by a total of 53 mm yr$^{-1}$. These changes occurred spatially relatively uniformly, implying that the relative changes were larger in the drier areas.

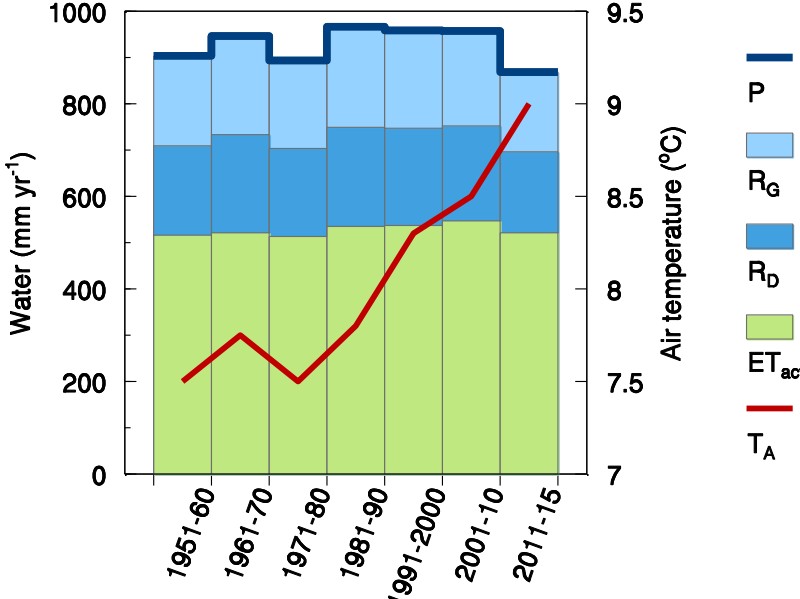

**Figure 1: Change in air temperature ($T_A$), falling precipitation (P), actual evapotranspiration ($ET_{act}$), direct runoff ($R_D$), and groundwater recharge ($R_G$) during the past seven decades and averaged over entire Bavaria (approx. area 71,000 km²); the data were compiled from Baumeister et al. (2017) who had applied a hydrological model (GWN-BW) particularly designed to calculate groundwater recharge. Like almost all hydrological models, the model considers only coarse land-use classes and does not account for the changes that occurred within the land uses. Furthermore, it does not consider lateral interactions between land uses (e.g., oasis and clothesline effects on evapotranspiration, run-on infiltration, and interflow). Hence, the model results almost entirely depict meteorological changes in space and time.**

Only a marginal decrease could be detected in summer rainfall between 1950 and 2020 (Fig. 2). Since 1990, the five driest summers (in ascending order: 1911, 1904, 1952, 2003, 1947) and the five wettest summers (in descending order: 1926, 1924, 1966, 1954, 1948) show no $CO_2$ related pattern. Winter precipitation has slightly increased since 1950. Due to the opposing trends of summer and winter and the unaltered spring and autumn rainfall, annual precipitation has hardly changed (BySMUV, 2021). Future climate projections for RCP 8.5 until 2050 also do not suggest alarming changes in summer rainfall (RCP is the

representative concentration pathway, and RCP 8.5 reflects business as usual; for RCP scenarios, see van Vuuren et al., 2011). Furthermore, in the latest analysis of return periods of heavy rainfall by the German Weather Service, covering 65 years until 2020, stationarity had still to be assumed for all 22 analyzed event durations spanning from 5 min to 7 d despite the application of sophisticated statistical tools to detect a $CO_2$-driven climate change signal (Haberlandt et al., 2023; Willems et al., 2023; Shehu et al., 2023). These minor changes in annual rainfall, seasonal, rainfall or event rainfall do not align with the severity of

floods and droughts experienced.

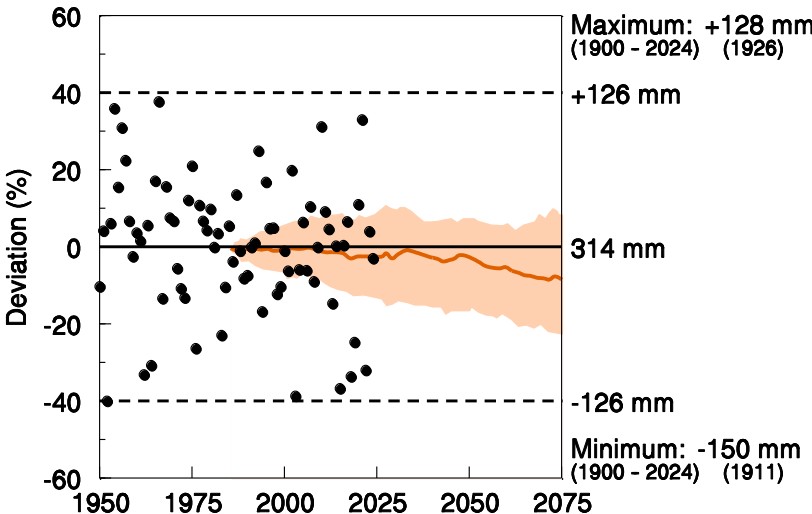

**Figure 2: Deviations of summer rainfall (June, July, August) in Bavaria (approx. area 71,000 km²) from the average**
**1971 to 2000 (being equal to the average 1900 to 2024); black circles and black curve show measurements, orange shaded area and orange curve show the bandwidth and the median of projections for RCP 8.5 (Data redrawn from BySMUV, 2021, and complemented with years 2020 to 2024 taken from DWD, 2025, which was also used by BySMUV, 2021).**

The only measured parameter except for air temperature, which already exhibits a $CO_2$-driven climate change signal, is annual rain erosivity (Fiener et al., 2013; Auerswald et al., 2019a). Erosivity is not a hydrological parameter but quantifies the ability of rains to detach soil particles and transport them. It is the product of rain amount and rain intensity of rain exceeding certain thresholds (Wischmeier, 1959; Wischmeier and Smith, 1958). As such, it is more sensitive to changes in cloud dynamics but also to the transition of snow to rain than the amount of rain, which is important for hydrology. Rainfall erosivity has doubled
in Germany since the 1960s (Auerswald et al., 2019a; b; Winterrath 2023), and projections for RCP 8.5 indicate further increases (Auerswald et al., 2019c; Fig. 3 top panel). Also, new convection-permitting climate simulations show a similar trend (Uber et al., 2023).

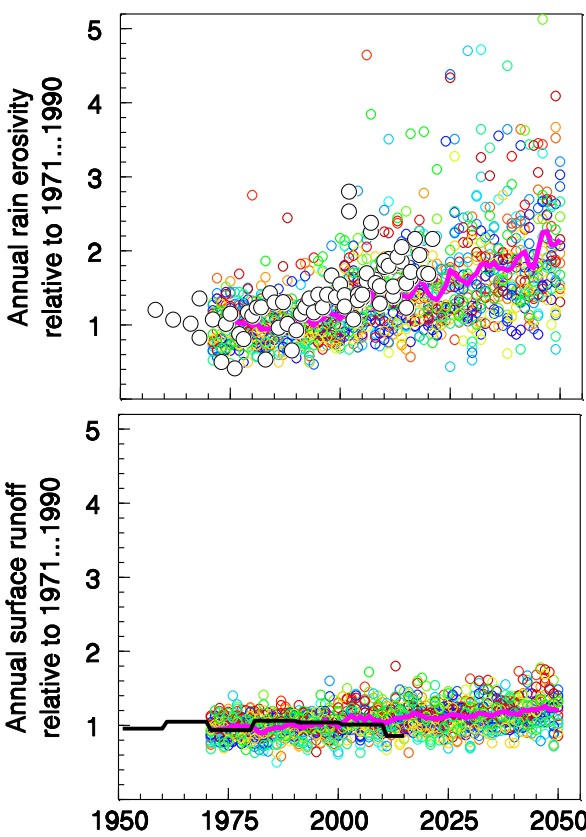

**Figure 3: Change in annual rain erosivity (top panel) and surface runoff (bottom panel) relative to the mean from 1971 to 1990. Colored symbols (taken from Auerswald et al., 2019c) were calculated from an ensemble of ten projections approved by the German Weather Service following RCP 8.5. Each colour indicates one of the ten climate projections. See the Appendix for the climate projections, methods, and covered area. The magenta line is a 3-year moving average of 10 projections (thus equivalent to a 30-year normal period). The black symbols (top panel) show the measured erosivities taken from (Auerswald et al., 2019a and Winterrath, 2023). The black line (bottom panel) shows the 10-year means of a full hydrologic model based on measured meteorological data (taken from Baumeister et al., 2017; for other hydrologic parameters resulting from this model, see Fig. 1).**

Thus, under climatic conditions typical for Central Europe until now, the $CO_2$-driven climate change has mainly caused an increase in rain erosivity eventually promoting soil crusting and, in turn, causing infiltration-excess runoff that contributes to flooding. Subsequently, drought may result because the runoff does not reach the soil store and, in the long-term, the soils' storage capacity is impaired due to increasing erosion (Fig. 4, left panel, shows the conceptual mechanism). This reduces evapotranspiration and temperature increases. The increasing temperature amplifies the drought: the so-called "event self-intensification" (Miralles et al., 2019). As the drier area heats up more than neighboring well-watered areas, the higher temperature and air of low humidity spreads to these nearby areas and causes them to transpire more until they also run short of water. Thus, the area with reduced evapotranspiration grows and may finally spread over an entire continent, described as "event self-propagation" by Miralles et al. (2019). Even though persistent large-scale circulation anomalies are critical for

the initiation of drought and heatwaves (Miralles et al., 2019), the length of such anomalies required to initiate droughts or heatwaves will depend on the moisture storage on the ground and whether it is sufficient to bridge the anomaly or not.

Modelling suggests that other mechanisms than $CO_2$-driven climate change must significantly contribute to floods and drought.
Studies, modelling surface runoff with the SCS curve number model (Woodward et al., 2002; NRCS, 2004) while assuming otherwise time-invariant soil and land-use conditions (see supplement) yielded a 20 % increase in yearly surface runoff by 2050 (Fig. 3 bottom panel), while for the same projections, erosivity is expected to increase by 100 % (Fig. 3 top panel). This expected runoff increase appears moderate compared to the interannual variation. Hydrological modeling using measured meteorological data between 1950 and 2015 confirms that changes in annual surface runoff driven by changes in
meteorological conditions are minor (Fig. 1) and do not support the notion that $CO_2$-driven climate change is the main cause of the increasing frequency of floods and droughts. Also, the $CO_2$-driven increase in evapotranspiration due to rising temperatures cannot explain droughts because evapotranspiration rises only modestly by 2 to 3 % $K^{-1}$ temperature increase (Lambert and Webb, 2008; Roderick et al., 2014; Bürger et al., 2014; Skliris et al., 2016). Therefore, a 2 K temperature rise would increase evapotranspiration by only 4 to 6 %, which should be buffered by functioning soils in humid areas.
Consequently, hydrological modeling indicated no increase in evapotranspiration during the last seven decades (Baumeister et al., 2017; Fig. 1). At present, $CO_2$-driven climate change's influence on the loss of water by surface runoff or on evapotranspiration does not explain why floods and droughts are already severe today.

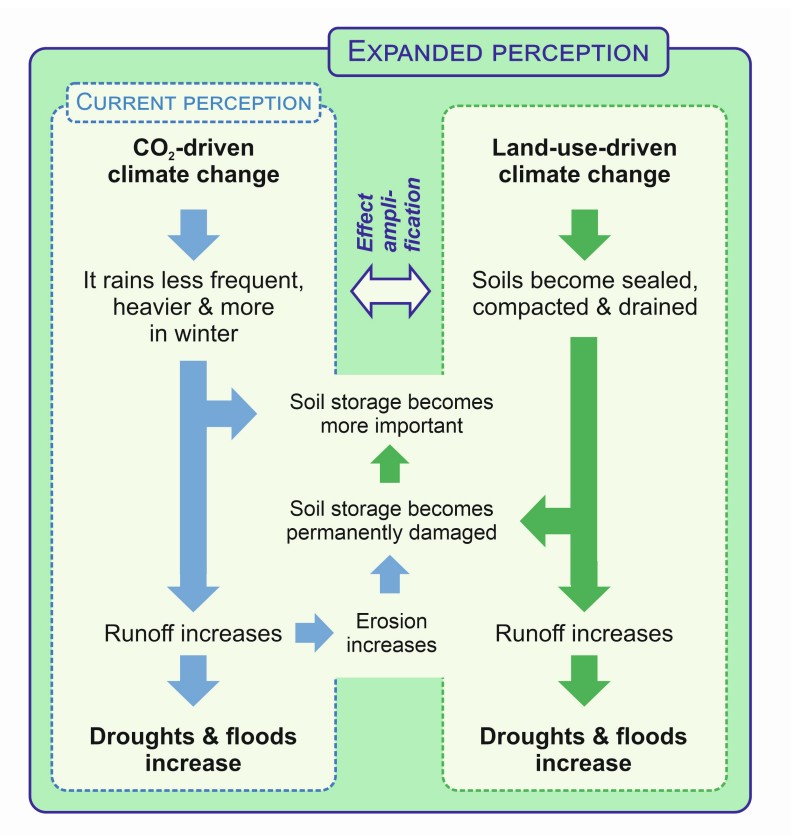

**Figure 4: Current perception of the influence of $CO_2$-driven climate change on rain and subsequent droughts and floods (left panel) and expanded perception, including land-use-driven climate change (right panel) and their interaction.**

## 3 Land-use-driven climate change

Land use has significantly changed over the past two centuries, and particularly since World War II. Many soils have been sealed by pavement or roofing; in addition, agricultural soils have been compacted, and drainage systems have been introduced. Sealing, compaction, and drainage lead to rapid water runoff, flooding, and, as less water enters the soil (Fig. 3, right panel). This decreases evapotranspiration and increases temperature. The effects appear almost identical to those induced by the $CO_2$-driven climate change. Thus, these effects can easily be misinterpreted regarding their causes and require closer examination.

### 3.1 Soil Sealing

Excluding farm and forest roads, each inhabitant in Bavaria is jointly responsible for sealing approximately 330 square meters of soil (Üreyen and Thiel, 2017), which, in total, accounts for about 6.0 % of the total land area. Although this percentage may not seem substantial at first glance, it has significant implications. Assuming all rainfall runs off from this area, an average

annual precipitation of 938 mm yr$^{-1}$ (Baumeister et al., 2017) means that, on average, 56 mm yr$^{-1}$ is directly converted to runoff equal to the mean precipitation in March. Notably, this precipitation loss exceeds what can be expected from $CO_2$-driven climate change (compare to Fig. 1). Except for the small amount of water left on sealed surfaces after a rain, sealed surfaces do not contribute to evaporation but partition their radiant energy uptake almost exclusively into sensible heat (Oke, 1982).

Thus, 6 % of the overall mean actual evapotranspiration of 528 mm yr$^{-1}$ (Baumeister et al., 2017) results in a calculated evaporation loss of 32 mm yr$^{-1}$. To put this into perspective, the energy required to evaporate 1 mm of water (1 L m$^{-2}$) can heat the atmosphere above 1 m² of ground approximately by 10 K to a height of around 200 m. Consequently, a loss of 32 mm of evaporation would theoretically lead to a 320 K increase in temperature across a 200 m high air column over Bavaria. However, this extreme scenario does not occur due to increased evapotranspiration from surrounding non-sealed areas (Zipper et al.,

2017), known as the oasis effect (Allen et al., 2000) or micro-oasis effect (Oke, 1982). Eddy diffusion or advection transfers energy from the sealed surface to adjoining areas (Calder, 1949; McNaughton, 1978, Klaassen and Claussen, 1995). Evapotranspiration in the surrounding of sealed surfaces thus can exceed even potential evapotranspiration due to the advection of warmer, drier air. This increase in evapotranspiration could be as much as 30 % (Oke, 1982) or even 50% (Panin et al., 1998) and be distributed over larger areas: in eddy covariance measurements, fetch areas of 100 times the instrument height

are often assumed, but this grossly underestimates travel distances (Leclerc and Thurtell, 1990; Savage et al., 1996). Travel distances of eddy diffusion determined with tracers can be larger than 3 km (Drivas and Shair, 1974), while theoretical considerations show that the advective exchange can be substantial to 20 km or more (McNaughton, 1976). On shorter distances, evapotranspiration caused by advection can even be 90 % of the total evapotranspiration in some cases (Prueger et al., 1996). However, the effects on evaporation must not necessarily be greatest near a dry surface because stomatal

conductivity decreases pronouncedly with increasing vapor pressure deficit (Grossiord et al., 2020). Thus, the compensation may extend over large distances due to reduced stomatal conductivity near the source of dry air (Baldocci and Rao, 1995). Even in rough vegetation like forests, where momentum fluxes decrease rapidly, heat fluxes extend over much larger distances (Klaassen et al., 2002). The reduced stomatal conductance under high vapor pressure deficit leads to a cascade of subsequent impacts, including reduced photosynthesis and growth, and higher risks of carbon starvation and hydraulic failure (Grossiord

et al., 2020). This may also explain why yields are often reduced along roads (Raatz et al., 2019).

Consequently, agricultural land and forests must provide additional water for evapotranspiration to compensate for this societal-induced increase in evaporative demand. It is known that the cooling effect of the vegetated area may extend up to 2 km into a built-up area (see Yan et al., 2018). Thus, it can safely be assumed that at least most of the sensible heat produced by sealed surfaces in the countryside, where sealed surfaces occur as narrow street bands, isolated buildings, or small villages,

is dissipated to the neighboring vegetated areas. The old references given above (e.g., Calder, 1949; McNaughton, 1978) show that we have known the effects of advection and lateral interaction for many decades, but except in combination with the urban heat island effect or eddy covariance measurements, they are hardly considered in landscape planning or hydrological modelling.

Soils with limited water storage capacity may not meet this additional demand for evapotranspiration, particularly during dry years. This will lead to reduced evapotranspiration from these soils, resulting in increased temperatures above these soils. The remaining area has to compensate even more advective energy resulting from reduced evapotranspiration. The area showing water shortage thus grows (event self-propagation). Additionally, it becomes warmer and the effect intensifies. A heatwave and a drought may result only because no action was taken to compensate for the adverse effects of soil sealing.

Natural systems also contain areas that generate runoff and may experience droughts. Typically, every landscape includes wet depressions where this runoff accumulates, acting as buffers during dry spells. Unfortunately, many of these wet areas have been systematically drained, and no compensatory wetlands have been established to offset the increased generation of runoff due to soil sealing. Soil sealing, mainly at the expense of cropland, created the pressure to drain wet grassland and convert it to cropland (Van der Ploeg et al., 1999; 2000; see Supplement for statistical data).

Furthermore, sealed areas impede groundwater recharge. Six percent sealing reduces the overall mean groundwater recharge (206 m $yr^{-1}$, Baumeister et al., 2017) by 12 mm $yr^{-1}$. Neighboring areas, if they compensate for the loss of 32 mm $yr^{-1}$ in evapotranspiration (Blumröder et al., 2021; Herbst et al., 2007), will, in consequence, recharge 32 mm $yr^{-1}$ less groundwater. Ultimately, this may lead to a calculated 44 mm $yr^{-1}$ decrease in groundwater recharge if vegetated surfaces compensate for the entire loss of evaporation caused by sealed surfaces. A loss of 6 % of the area by sealing thus could potentially convert to a loss of 21 % in groundwater recharge. This agrees with the declining water tables observed in many aquifers. Between 2000 and 2020, approximately 20 % of the 1600 monitored aquifers in Bavaria experienced a significant decline in water levels, while another 20 % declined slightly (Bayer et al., 2022).

### 3.2 Drainage

Landscape hydrology (Fig. 5) comprises the coupling of vertical fluxes (precipitation, infiltration, evapotranspiration, groundwater recharge) with lateral fluxes (surface and subsurface runoff, groundwater flow, air moisture transport). Although frequently neglected or overlooked in hydrological models, the lateral fluxes are crucial for exchanging water within a landscape (Arnault et al., 2021). Surface runoff often infiltrates while traversing the landscape (run-on infiltration; Woolhiser et al., 1996), contributing to groundwater recharge (Carroll et al., 2019; Fiener and Auerswald, 2003). Subsurface flow (interflow) laterally percolates through the soil, supplying water to lower slopes for extended periods without rain, eventually contributing to groundwater recharge (Carroll et al., 2019). This enables lower slopes to continue evaporating, even in rainless periods, increasing air humidity and thus alleviating water stress also in upslope areas. Groundwater flow sustains riparian areas and rivers and both enhance air humidity and act as buffers (Auerswald et al., 2019 d).

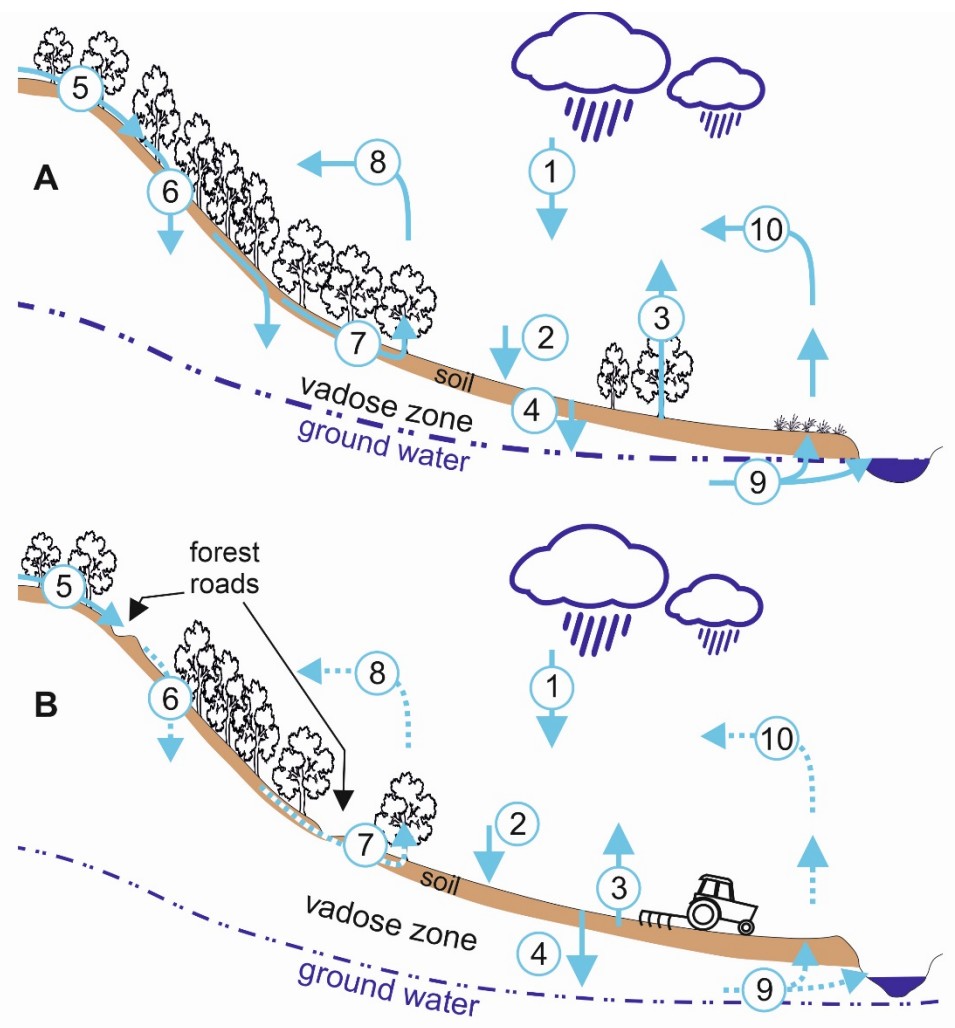

**Figure 5: Schematic representation of (A) intact landscape hydrology and (B) under contemporary conditions where lateral water flows have been strongly disturbed (dashed lines) by infrastructure and other interventions and where the groundwater was lowered (1 precipitation, 2 infiltration, 3 evapotranspiration, 4 groundwater recharge, 5 surface runoff, 6 run-on infiltration, 7 interflow, 8 air-moisture transport, 9 groundwater flow, 10 groundwater-born evapotranspiration and air-moisture transport).**

These buffering fluxes, vital for reducing local climatic extremes (Ripl, 2003), have been substantially reduced. This reduction may stem from limited awareness of their existence and the features that enhance and restrict them. The creation of field and property boundaries may restrict fluxes between neighboring lands. For example, roadside ditches form a dense drainage network today. While the total stream length in Bavaria is 100000 km (1.42 km km$^{-2}$; LfU, 2024), the length of public roads is 141800 km (2.01 km km$^{-2}$; ByStMWBV, 2018), and the length of farm roads is 200000 km (2.83 km km$^{-2}$; Anonym 2018). Due to construction regulations (FGSV, 2021), roads are usually accompanied by roadside ditches on one or both sides. Thus, the drainage network created by roadside ditches is three to six times as long as the natural drainage network. It collects runoff

over short distances, preventing it from entering neighboring fields with available infiltration capacity. Run-on infiltration, a valuable process in heterogeneous landscapes, is often hindered (Fiener et al., 2011), and groundwater recharge is reduced, and as runoff is now often directly and effectively transferred to river courses (Fig. 5) meaning there is limited scope for the retention of sediment and other pollutants.

Furthermore, runoff is accelerated and conveyed downstream alongside the road to the next settlement, increasing the risk of flooding. The flow velocity in ditches can be up to 20 times greater than in shallow runoff across fields and grasslands (typical flow velocity on fields: ~0.1 m s$^{-1}$, while roadside ditches may reach 2 m s$^{-1}$; see Seibert and Auerswald, 2020). Peak runoff directly correlates with flow velocity, exacerbating flood risks (Gericke and Smithers, 2014). The homogenization of landscapes further amplifies the issue. Thus, flooding the next village becomes more likely (Bronstert et al., 2018). The fact

that surface runoff is generated, which cannot be avoided entirely, even in natural systems, is not the main problem. The problem was amplified by creating an efficient runoff drainage system through ditches and pipes and homogenizing landscapes that amplifies peak runoff even at constant runoff volume.

In addition to surface runoff, subsurface runoff is also conveyed by drainage systems. Even in the driest regions of Bavaria, comprehensive subsurface drainage via tiles was carried out with government support, which is well documented in some

regions (Fig. 6). It is challenging to envision significant groundwater recharge occurring under such extensive tile drainage systems. Drainage will not only affect the drained areas but can also cause higher water losses due to evapotranspiration driven by advection on the neighboring undrained areas (Baldocchi et al., 2016, Klaassen and Claussen, 1995). According to Tetzlaff et al. (2010), 23% of the agricultural land in Germany has been artificially drained, and drainage runoff in the southern part of Bavaria can be up to 500 mm yr $^{-1}$ (Wolters et al., 2023). In consequence, after subtracting the water lost by artificial drainage,

the remaining precipitation in the south, which has the highest rainfall in Bavaria, is as low as the total precipitation in northern Bavaria where the rainfall is lowest.

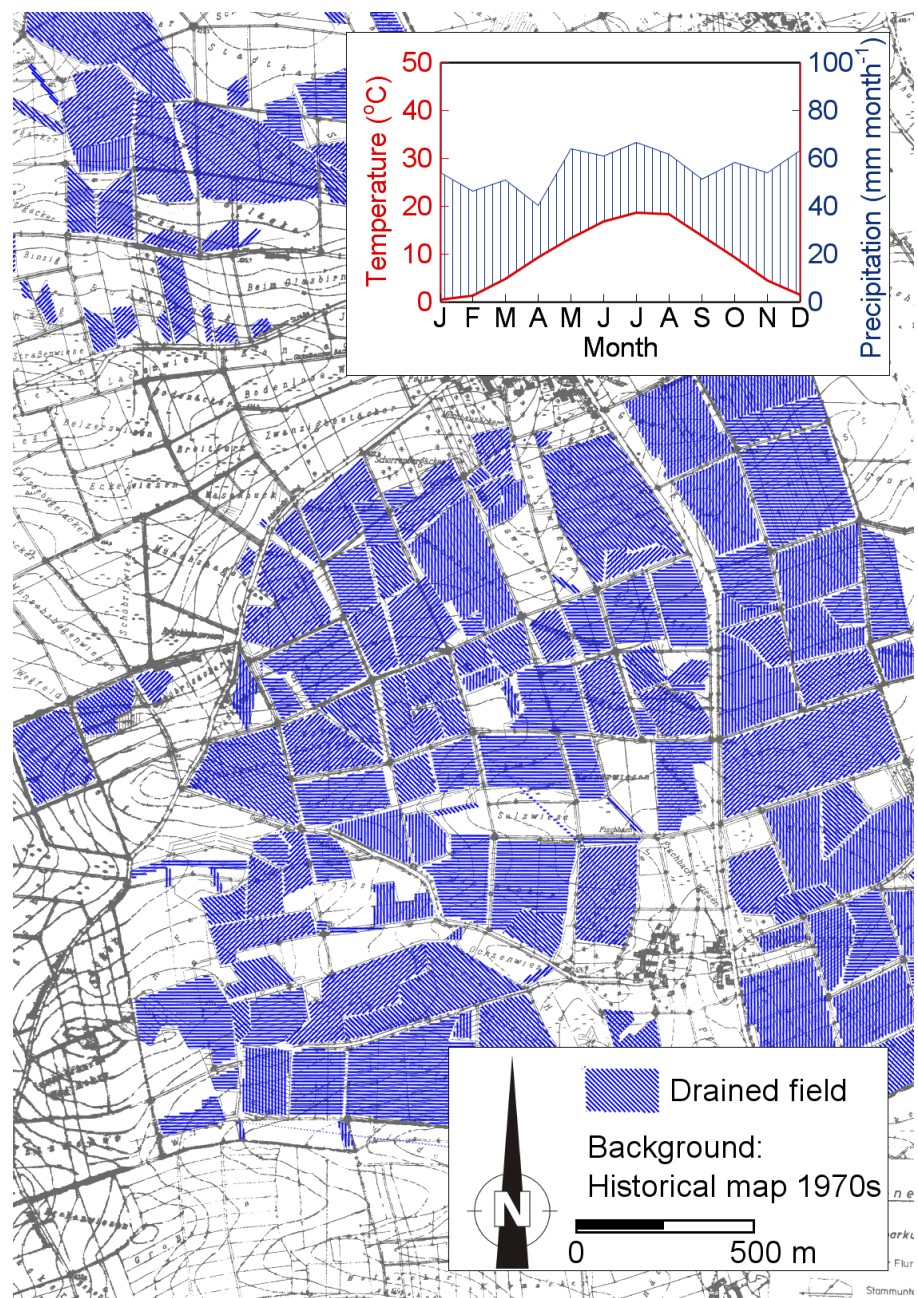

**Figure 6: Extensive application of tile drainage (blue-shaded fields) can be found even in landscapes with rainfall far below the Bavarian average. The inset is a Walter-Lieth climate diagram for the meteorologic normal period 1991 to**
**2020 (mean annual precipitation 670 mm yr⁻¹). The background shows the original plan from the 1970s for the water association of Oberschreckenbach for the village of Gumpenhofen, county of Rothenburg ob der Tauber, Middle Frankonia, on which tile lines were redrawn in blue.**

Artificial drainage is not limited to agricultural land; forests have also been drained. These projects, some dating back over a century, are often forgotten, yet the drains remain effective (Tempel, 2006). In recent decades, the extensive network of truckable forest roads has initiated an unintentional forest-wide drainage system. Running perpendicular to the main slope, these roads deeply cut into slopes. As forest soils are prone to high subsurface runoff due to their high infiltration capacity, but contrasting hydraulic conductivities between different horizons (Hümann, 2012; Nordmann, 2011) exfiltration at roadcuts can occur. This exfiltrated runoff is then drained via road ditches and bypasses the area downslope of the road (Wigmosta et al., 2002). The mean density of truckable forest roads in Germany is 4.5 km km$^{-2}$ (BMEL, 2021), restricting surface runoff to short distances (on average 220 m, but shorter in steep terrain).

Another cause of additional drainage is the lowering of the groundwater tables. This made drainage by tiles and ditches possible in some areas, but it also led to a decoupling of the rivers and their riparian areas (Auerswald et al., 2019d; Fig. 4) and caused many small rivers to disappear (Reckendörfer et al., 2013, Zerbe, 2013) while the remaining increasingly suffer from drought and heatwaves. This poses a major threat to biodiversity (Drainas et al., 2023, Cushway et al., 2024). Riparian areas connected to the groundwater serve as sources of air moisture even in prolonged drought periods, while after groundwater lowering coarse-textured riparian areas, associated with river gravel deposits, dry out first and then may act as nucleus of an expanding drought. This effect of dry riparian areas is accelerated when river courses are straightened, leading to increasing transport capacity and hence initiates river incision.

### 3.3 Soil Compaction

The weight of agricultural and forestry machinery has steadily increased since World War II. This upward trend continues unabated. For instance, the wheel load of grain harvesters available in a specific year increased linearly from 2 t to 7 t between 1960 and 2000 with no discernible slowing of the trend (Keller et al., 2019). While this increase in weight was offset at the soil surface by expanding the tire contact area, maintaining nearly constant contact-area pressure, subsoil compaction increased (Keller and Or, 2022). Subsoil compaction is primarily determined by wheel load, becoming largely unavoidable if the wheel load exceeds 3 to 5 t (Soane, 1983; Håkansson et al., 1994; Schjønning et al., 2012). Since the 1990s, such wheel loads have been common in all major agricultural machinery (Keller and Or, 2022). In a recent (2011-2018) German-wide survey along an 8 km × 8 km grid analyzing 16778 soil samples, 51 % of the arable land had root restrictions caused by compaction (Schneider and Don, 2019). This value is similar to the 43 % of over-compacted soils found in the Netherlands (Brus and van den Akker, 2018).

Remarkably, the rapid increase in agricultural yields observed between 1880 and 1990, culminating in a fivefold increase (Mahlerwein, 2016), with the rise being fastest between 1960 and 1990, suddenly stopped in 1990 when wheel loads of major agricultural machinery exceeded 5 t. Despite significant advancements in plant breeding (Guarin et al., 2022), no further yield increases were achieved for three decades. This is observed also in other countries that had similar developments in agricultural machinery weights (Keller et al., 2019).

The stagnation in crop yields can be attributed, at least in part, to the slowdown in root growth even under slight subsoil compaction (Bengough et al., 2011). Upon tillering, gramineous crops produce nodal roots starting close to the soil surface; similarly, legumes produce basal roots (Tajima, 2011). These shoot-borne roots dominate water uptake (Krassovsky, 1926). Based on measured bulk densities, root growth models indicate that it took roots two to three weeks to reach a depth of 50 cm in the 1960s. Today, it takes over two months to cover the same depth (Keller et al., 2019). Consequently, restricted subsoil exploration forces plants to extract all their water needs from the topsoil. This situation mimics meteorological drought conditions because water depletion of the topsoil increases temperatures and reduces air humidity, although it is of physiological origin. This likely results in a misleading interpretation of the origin of the drought.

However, compaction might not only result in a higher drought risk but also impede water percolation and increase the susceptibility to waterlogging (Hartmann et al., 2012). The unforeseen extreme wheat failure in France in 2016, which even exceeded the yield loss during the centennial drought in 2003, was mainly caused by anoxia during a cool and wet May. Such conditions are expected to become more frequent due to $CO_2$-driven climate change (Ben-Ari et al., 2018; Nóia Júnior et al., 2023). Impeded percolation by subsoil compaction also causes saturation-excess surface runoff and soil loss when the soil above the compacted layer becomes saturated (Verbist et al., 2007).

## 3.4 Hedgerows

Hedgerows, which owe their existence to agriculture, have largely been lost in Bavaria, like elsewhere in Western Europe (Forman and Baudry, 1984; Meeus, 1993). The positive impact of hedgerows on agricultural yields has long been recognized (e.g., Wendt, 1951) and is supported by numerous studies (Sudmeyer et al., 2007; Veste et al., 2020). The primary mechanism involves reducing wind velocity close to the soil surface, which decreases evapotranspiration. Calculations specific to eastern Germany suggest that evapotranspiration can be reduced by nearly 100 mm $yr^{-1}$ over a distance equivalent to 25 times the height of the hedge (Funk et al., 2022). This compensation effectively counteracts increasing fluctuations in precipitation due to $CO_2$ effects on climate.

Furthermore, hedgerows influence the diurnal variation in air temperature (Forman and Baudry, 1984). Nights become cooler, reducing plant respiration (Ryan 1991) and promoting dew formation (Monteith, 1957). Conversely, daytime temperatures rise, enhancing quantum yield efficiency and assimilate gain (Ehleringer and Bjorkman, 1977). Despite these benefits, many hedgerows have been removed. This may be attributed to a misinterpretation of yield gains, which are maximized at a distance roughly five times the hedge height. The diminishing yield as distance to the hedge decreases can be misconstrued as yield loss caused by the hedge, while it actually represents diminishing yield gains.

## 4 Scale dependence

Land-use impacts, such as wheel tracks or soil sealing, occur on a much smaller spatial scale than meteorological phenomena like rain cells or atmospheric pressure systems. However, land-use practices also affect vast areas due to the widespread

standardization of methods, regional specialization, and the increasing size of agricultural fields. Wheel tracks, although narrow compared to meteorological impacts, typically cover entire fields. Analysis indicates that fields experience 10 to 20 wheel passes per year leaving almost no gaps between wheeled areas. Less than 10% of the field area are wheeled less frequently and around 10% wheeled more often (Augustin et al., 2020). Given that wheeling patterns vary with crop type, the

persistence of subsoil compaction, and that the load is supported by an increasing area with increasing depth, a largely uniform and comprehensive coverage by wheel tracks can be assumed for a crop rotation. Furthermore, similar agricultural machinery is employed across industrialized nations. The increasing weight of machinery is not confined to specific crops or production systems but affects cropland, grassland (Horn et al., 2020), and forested areas (Cambi et al., 2015), impacting both conventional and organic farms. This trend is driven by market forces that influence all forms of production, leading manufacturers to scale

up the size of all types of machinery (Keller et al., 2019). The differences are thus only gradual or due to the use of older machinery. Given that these mechanisms have been in operation for decades, and that critical wheel loads exceeding 5 tons have been surpassed since the 1990s, it is reasonable to assume that widespread exposure to high wheel loads has led to extensive subsoil compaction.

A similar argument applies to other localized impacts like soil sealing. The focus in the scientific literature on the urban heat

island effect can create a misleading impression, stemming from an anthropocentric perspective and the concentration of humans in urban areas. From a hydrological perspective, soil sealing is more problematic in rural areas, where more roads and infrastructure are required to support a smaller population density, and houses have fewer floors than in cities. As a result, the sealed areas per inhabitant is up to 40 times larger in rural counties than in metropolitan counties even without consideration of farm and forest roads (Üreyen and Thiel, 2017). Thus, land-use impacts like subsoil compaction or sealing can affect areas

comparable to or even larger than atmospheric pressure systems.

A fundamental assumption in field-scale and farm-scale experiments is that farmers have the capacity to implement improved land management practices. However, this perspective requires reconsideration in light of global market mechanisms and the large-scale and overarching changes, such as the increasing weight of agricultural and forestry machinery. Even the most conscientious farmer will be unable to buy light machinery if it is unavailable in the market (Keller et al., 2019). Furthermore,

farmers may often remain unaware of these limitations due to a lack of unbiased, industry-independent guidance (Schnyder et al., 2019). As the scale of these impacts grows, so does the need for expanded institutional accountability.

Rain cells or pressure systems move over a location within hours or days, subsoil compaction can persist for years or even centuries, tile drainage can remain functional for many decades or longer, and soil sealing is rarely reversed. Hence, on a temporal scale, these influences of land use last considerably longer than meteorological phenomena.

Furthermore, the pace of land-use change is far more rapid than changes in atmospheric $CO_2$ concentration. While it has taken nearly three centuries to increase pre-industrial $CO_2$ levels by 50% (https://climate.nasa.gov/vital-signs/carbon-dioxide), the weight of grain harvesters doubled in just a decade between 1960 and 1970 (Keller et al., 2019). As a result, we have had much more time to anticipate and understand the effects of $CO_2$, while the large-scale effects of rapid land-use change remain poorly understood. Current research tends to focus on the short-term, localized impacts of land-use changes, such as the effects of

soil compaction on crop yield or water loss in individual fields. However, the broader, synchronized changes in land use across regions and continents, which affect much larger areas than individual fields, are only beginning to be explored.

## 5 Remedies

Land use intervenes in multiple ways with the water and energy budget. This should allow, at least in a temperate climate, to
cushion the adverse effects of the $CO_2$-driven climate change and to compensate for the adverse effects of land-use changes of the past.

Sealing is the most significant intervention in soil functioning. This refers not only to cities where the urban heat island effect directly affects humans. Cities are responsible only for a small yet concentrated share of the total sealed surface. The five largest cities in Bavaria contribute only 10 % to the total sealed surface (Esch et al., 2007), illustrating the importance of sealed
surfaces for peri-urban and rural areas. Action against sealing is urgently required. Possibilities are manifold. They include unsealing paved surfaces (e.g., parking lots), installing photovoltaics above sealed surfaces to remove some of the solar energy as electricity instead of converting it into latent and sensible heat, and greening (green roofs, tree alleys). Since technical measures usually replace only one soil function at a time, combining measures may be necessary (simultaneous unsealing, greening, and photovoltaic on parking lots). In addition to sponge cities, sponge towns, sponge villages, and sponge landscapes
are required. As the effect of a sealed surface on the water and energy budget is universal, the remedies will also have to be universal. Hence, the responsibility rests on everybody, not only a few city planners.

The landscape requires more hedges or structures similar to hedges. Again, many measures are available, such as solar fences, agroforestry, or tree alleys, all of which can reduce wind velocity. In particular, heavy-traffic roads, which cannot be unsealed, should be accompanied by hedges or tree rows to mitigate their climate-adverse effects. This insight was already gained by
Napoleon Bonaparte 200 years ago, who let trees be planted along roads to improve the microclimate for his marching soldiers (Balmer, 2022).

In agriculture, two requirements are most important: lowering wheel loads and improving soil cover by living or dead plant material. These will directly affect soil functioning regarding water and energy balances and increase C storage in soil/ contribute to $CO_2$ sequestration. Both requirements can be met with many economically advantageous solutions (e.g.,
Auerswald et al., 2000). The most important obstacle to their adaptation appears to be the lack of industry-independent advice to farmers (Schnyder et al., 2020).

Irrigation appears not to be an eligible remedy as it may reduce water shortages in the irrigated field, but it could not solve anoxia or runoff problems; instead, it increases their likelihood. The damage is usually greater than the benefit. Root zone storage capacity decreases with irrigation (van Oorschot et al., 2024) elevating water demand, and water storage capacity may
decrease due to irrigation-induced erosion (Batista et al., 2023). Irrigation will always increase water consumption and, in turn,

water scarcity. The only exception is irrigation using water that cannot be infiltrated, e.g., from sealed surfaces, that is stored and used in dry periods (example: https://www.vin-aqua.de/).

A climate-friendly land use is possible. However, changes are required in so many places that governmental measures like laws or subsidies can never achieve this. Instead, a paradigm change is necessary. The old food security paradigm is subordinate now because it can only be accomplished with climate resilience. The old paradigm of economic efficiency is outdated because efficiency and resilience are mutually exclusive.

## 6 Conclusions

Undoubtedly, measures against climate change by reducing $CO_2$ emissions are essential and have become a global policy target. However, exclusively focusing on this goal ignores other important mitigation measures that urgently need to be realized. As illustrated here, restoring hydrologically functional landscapes and soils should be considered equally important to mitigate climate change, especially concerning extremes such as floods, droughts, and heatwaves, and to preserve the foundation of food and life. The question of whether land use or $CO_2$ are more significant drivers of floods and drought deserves more attention even though a simple answer will never be possible.

Measures of reducing soil sealing and compaction and retaining water in structurally rich landscapes (sponge landscapes and sponge cities) can all have pronounced climatic effects. Since they also comprise quick and simple measures (e.g., not incorporating straw after the grain harvest), which deliver a measurable cooling of a few K, they offer a greater level of acceptance by the public compared to measures requiring personal lifestyle changes. Policymakers and planners should, therefore, emphasize soil functioning and water retention in climate change policy action.

The most significant challenge may rest on hydrological science, where we largely neglect lateral interactions happening in the atmosphere, on soil surfaces, and in soils. We poorly address lateral phenomena like advection in the atmosphere, run-on infiltration, or subsurface flows. Physical experiments designed to analyze the influence of lateral interactions on the landscape scale are almost impossible to conduct, as extensive areas would have to be included, manipulated, and replicated to fulfill statistical requirements. Hence, we rely on modeling. However, also our models often disregard these lateral effects. Field size and neighborhood hardly play a role in many model calculations like evaporation. Land use is typically considered in broad categories like forestland, grassland, cropland, and urban land. We use parameter values derived decades ago, which hardly reflect the unprecedented changes within each land-use category during the last decades. For instance, no hydrological model requires agricultural machinery weight and can reflect its effects. Soil properties are treated as constants in most hydrological models, although we know from a multitude of experiments that land use modifies them. Meteorological parameters, such as temperature, humidity, wind, and rainfall, are employed as external controls even though they are influenced by the system, thereby engendering feedback mechanisms. We use models that consider a strongly limited number of controls to find, in circular reasoning, that these controls are important. Moreover, most models use similar or even identical controls instead of complementing each other.

Even the presently strongest $CO_2$-driven climate change effect in rural areas except for the temperature increase, the pronounced increase in rain erosivity, which causes soil crusting and infiltration-excess runoff, is hardly captured by hydrological models. Most of them only implement percolation-excess runoff (e.g., via the Green-and-Ampt approach) but do not consider the drivers of crusting.

Comprehensively considering all effects in modeling is hindered by data limitations, computational time constraints, and the unfavorable behavior of models that consider feedback mechanisms such as increasing complexity, instability and exponential growth or oscillations. Consequently, it is crucial to acknowledge our limited understanding of land-use effects. Any conclusions regarding the impact of land use based on modeling must be drawn cautiously, regardless of the apparent certainty of modeling results. In turn, considering meteorological changes only in the light of greenhouse gasses is biased by the same limitations.

**Author contributions**

KA led the conceptualisation of the paper and led the writing and drafted the figures, PF contributed to the figures and to the writing of the paper, JG and JQ contributed to the writing of the paper.

**Competing interests**

The authors declare they have no competing interests.

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
