# Peer review of "HESS Opinion: Floods and droughts - Are land use, soil management, and landscape hydrology more significant drivers than increasing CO2?"

_EGUsphere, 2024_

## Author Comment (AC1)

**Supplement to**
**HESS Opinion: Floods and droughts - Land use, soil management, and landscape hydrology are more significant drivers than increasing temperatures**

Karl Auerswald[1], Juergen Geist[1], John N. Quinton[2], Peter Fiener[3]

[1]School of Life Sciences, Technical University of Munich, Freising, 85354, Germany
[2]Lancaster Environment Centre, Lancaster University, Lancaster, LA1 4YQ, UK
[3]Working Group Water and Soil Resource Research, University of Augsburg, Augsburg, 86159, Germany

*Correspondence to*: John Quinton (j.quinton@lancaster.ac.uk)

**1. Description of the example area**

**1.1 Landscapes**

Bavaria, a federal state in the south of Germany, is 70550 km² in size and larger than about half of the countries in the European Union e.g. the Netherlands or Ireland (BStMWi, 2023). Geologically and geomorphologically, it can be separated into four zones (Ahnert, 1989). (1) In the south, there is the narrow strip of the German Alps, with the highest elevation being 2962 m a.s.l. (2) Adjoining to the north is the hilly Alpine Foreland. It ends at the River Danube, which crosses Bavaria from the west to the east. (3) North of the Danube is the cuesta landscape of the Mesozoic Scarpland. Most of this area drains to the River Main, a tributary of the River Rhine. The lowest point of Bavaria can be found in the north-west where the Main leaves Bavaria (about 100 m a.s.l.). (4) In the east, Bavaria is bordered by crystalline mountain ridges that are part of the Bohemian Massif (highest elevation 1458 m a.s.l.).

Loess-influenced soils are widespread except in the Bohemian mountain ridge and the Bavarian Alps. The loess deposits can be several meters deep in the east along the River Danube and in the north of the Mesozoic Scarpland. However, more commonly, the loess cover has a depth of one meter or less, making soil fertility vulnerable to soil losses by erosion.

**1.2 Climate**

The Köppen-Geiger climate zones change from Cfb (temperate oceanic) in the northwest to Dfb (humid continental with warm summer) in the south. Dfc (continental subarctic climate) or ET (tundra climate) can be found at high altitudes.

The weather and hydrological data, covering the period 1951 to 2015, were taken from Baumeister et al. (2017), who applied a hydrological model (GWN-BW) that is particularly designed to calculate groundwater recharge. Like almost all hydrological models, the model considers only coarse land-use classes and not accounting for the changes that occurred within the land uses. Furthermore, it does not consider lateral interactions between land uses (e.g., oasis and clothesline effects on evapotranspiration, run-on infiltration, and interflow). Hence, the model results almost entirely depict meteorological changes in space and time.

Annual rainfall in Bavaria increases from about 600 mm yr$^{-1}$ in the north to about 1400 mm yr$^{-1}$ in the Bohemian Massif in the east and to more than 2000 mm yr$^{-1}$ in the Alps in southern Bavaria. The overall spatio-temporal mean for Bavaria is 938 mm yr$^{-1}$.

Actual evapotranspiration in Bavaria is approximately 500 mm yr$^{-1}$ (overall mean 528 mm yr$^{-1}$), with higher values south of the Danube (mainly around 600 mm yr$^{-1}$), where sunshine duration is more prolonged (Klämt, 2008). In forests, evaporation in the south can even be around 700 mm yr$^{-1}$.

Total runoff (direct runoff and groundwater recharge; overall mean 409 mm yr$^{-1}$) reflects the pronounced gradient in precipitation, given that the variation in evapotranspiration is comparably small. For most of the country, total runoff varies between 200 and 400 mm yr$^{-1}$ but decreases to 100 to 200 mm yr$^{-1}$ in lowland areas that receive less rainfall (rain shadow) and experience higher temperatures and higher radiation (foehn effect). In particular, this is the case near the river Main and also in a 20 to 50 km wide belt south of the River Danube. High total runoff can be expected at high elevations, in particular in the Alps, where total runoff may even exceed 1500 mm yr$^{-1}$.

About half of the total runoff is groundwater recharge (overall mean 206 m yr$^{-1}$), and the other is direct runoff (overall mean 204 mm yr$^{-1}$). Both components have a similar spatial pattern as total runoff and vary between 50 and 600 mm yr$^{-1}$ depending on the region. A distinct deviation from this 50:50 ratio can only be found in the karst areas of the Mesozoic Scarpland, where a larger fraction of total runoff is allocated to groundwater.

The changes during the last seven decades have been pronounced for air temperature but minor for other meteorological and hydrological variables (Fig. A1). Annual mean temperature has risen from about 7.5°C to 9°C. Temperatures started to change circa 1980 and increased almost linearly until 2015. Spatially, this increase was uniform.

The temporal changes in precipitation, evapotranspiration, direct runoff, and groundwater recharge were minor compared to the spatial variation and without a clear trend. Between 1950 and 2015, groundwater recharge decreased by a total of 22 mm yr$^{-1}$ (i.e., on average, it decreased by 0.33 mm yr$^{-1}$ every year during the last 65 years), and actual evapotranspiration decreased by a total of 7 mm yr$^{-1}$, while precipitation decreased by a total of 53 mm yr$^{-1}$. These changes occurred spatially relatively uniformly, implying that the relative changes were larger in the drier areas.

[Figure]

**Figure A1: Change in air temperature ($T_A$), falling precipitation (P), evapotranspiration (actual $ET_{act}$), direct runoff ($R_D$), and groundwater recharge ($R_G$) during the past seven decades and averaged over entire Bavaria (data compiled from Baumeister et al., 2017).**

**1.3 Land use**

Population density is 190 km$^{-2}$ (total population: 13.4 million; BStMWi, 2023). Settlement and traffic area covers 12%, while forests contribute 37%, farmland 43%, and water bodies 2% (the missing area is unused land, mainly in the Alps, and other miscellaneous parts). Two-thirds of the farmland is cropland; one-third is grassland. Grain crops contribute 53% to the cropland, 29% are fodder crops (mainly silage maize), and 5% are row crops (except silage maize). The most important crop is winter wheat, followed by silage maize. About 30% of the agricultural area is cropped by farms 50 to 100 ha in size. About 50 % of the farmland is leased (Bayerisches Landesamt für Statistik, 2023).

The changes in land use during the last seven decades (Fig. A2) falsely indicate that forests and cropland hardly changed in terms of their percentage contribution, yet, grassland has decreased, and settlement and traffic areas continuously increased (Fig. A2). In reality, forests have become heavily fragmented, particularly by infrastructure projects. Although the lost forest area has been compensated at the expense of cropland and grassland, the forests have become cut in small pieces (for an example see Auerswald et al., 2019a), changing their hydrological behavior (increasing the clothesline effect, increasing drainage by roadside ditches).

Cropland has experienced the most significant losses. In particular, the best loessal soils and the flat areas were preferred construction sites. The hydrologic change is thus more pronounced than the aerial change because the loss predominantly occurred on soils with high water storage capacity and low surface runoff.

The loss of cropland is not evident from the overall numbers illustrated in Fig. A2 because it was compensated by converting grassland into cropland. This loss of grassland did not happen evenly but predominantly affected the grassland interspersed in cropland. These were wet depressions that were drained, riparian grassland where the groundwater was lowered, or the

grassland previously occurring along thalwegs (nowadays called grassed waterways when established by technical engineering). This interspersed grassland had traditionally been the fodder base for mixed farms. Thus, alongside the loss of grassland, an increase in fodder production on cropland occurred, namely silage maize. Note that the loss of grassland is almost perfectly mirrored by the rise in maize (Fig. A2). The loss of the interspersed grassland, combined with replacing organic with mineral fertilizers, allowed specialization of farms either focusing on animal production or cash crops. Again, this loss of grassland was hydrologically more critical than the loss of aerial share suggests because the wet places in the landscape became drained. Also, the areas predominantly contributing to run-on infiltration, the grassed thalwegs, disappeared. Increasing maize acreage promoted runoff, erosion, and soil evaporation in early summer, when maize cover is still sparse.

The loss of grassland has stopped in recent years because the conversion of grassland to cropland has been forbidden since 2019, although its conversion to settlement area would still be allowed. This conversion ban has caused the cropland to decrease since then because the losses of cropland to urban development cannot be compensated anymore by grassland conversion. This also illustrates that urban development predominantly ocured on cropland.

[Figure]

**Figure A2: Contribution of forests, cropland, grassland, and maize to total land in Bavaria during the past six decades. The maize area is included in the cropland area. The data were compiled from different official sources. The settlement area is displayed in Fig. A3.**

The areas of settlements and roads more than doubled during the past five decades (Fig. A3) and at a rate which is increasing. The sealing ratio, i.e., the fraction of the settlement and traffic area that cannot infiltrate anymore, is also increasing. The increase in the sealing ratio applies not only to the new settlement and traffic areas but to the total area covered by buildings and roads. This indicates that sealing continues on old settlement and traffic areas, e.g. by increasing the building density and sealing the open spaces to create parking lots.

[Figure]

**Figure A3: Increase in settlement and traffic area in Bavaria during the past five decades and sealing ratio of the settlement and traffic area. The data on settlement area were mainly taken from https://www.stmuv.bayern.de/themen/boden/flaechensparen/daten.htm; those on the sealing ratio were adopted from https://www.lfu.bayern.de/umweltkommunal/flaechenmanagement/bodenversiegelung/index.htm.**

**2 Prediction of surface runoff and rain erosivity from climate projections for Bavaria**

We used an ensemble of 10 climate projections from the EURO-CORDEX initiative (Jones et al., 2011; Knist et al., 2017) for the period 1970 to 2050 based on the RCP 8.5 scenario reflecting business as usual (Table A1). These projections were approved by the Deutscher Wetterdienst (German Weather Service) (Hübener et al., 2017). The data had a resolution of 0.11° (approximately 12.5 km), and they covered Bavaria, the hydrological catchment of Bavarian rivers, and a 50 km buffer area, thus extending to substantial parts of Austria, Switzerland, Italy, and the Czech Republic. The climate projections delivered daily air temperatures and daily precipitation.

Predicting runoff from daily precipitation was straightforward using the SCS curve number model (Woodward et al., 2002; NRCS, 2004). According to present land use, curve numbers for different crops, land uses, and hydrologic soil groups were taken from Seibert and Auerswald (2020). They were kept constant for the projection period to isolate precipitation effects from other effects. Any errors in the assumption of curve numbers will have a minor impact because we were only interested in the relative change of runoff, not its absolute amount. Nevertheless, Seibert and Auerswald (2020) have shown that these curve numbers adequately reflected runoff of 1174 events from 22 catchments in Bavaria ranging from 12 to 170 km² in size. The calculation of rain erosivity, which depends on rain amount and rain intensity, requires higher resolved rainfall than the daily rainfall available from climate projections. Considering the pronounced influence of air temperature on precipitation rate (Lenderink and van Meijgaard, 2008), a strong correlation between event erosivity, calculated from highly resolved rain gauge data, and daily air temperature and daily precipitation could be established from 605738 erosive events and validated by another 15133 events from an independently measured data set (erosivities of both data sets were taken from Fischer et al., 2018;

Figure A4). The correlation yielded the following transfer function (Eq. 1) between event erosivity (called erosion index $EI$ in N h$^{-1}$), daily precipitation ($P_d$ in mm d$^{-1}$) and daily mean air temperature ($T_a$ in degree Celsius).

$$EI = 1.57 \times 10^{-6} \times P_d^{2.63} \times T_a^{2.52} \tag{1}$$

135 Training data and testing data used event erosivities because event erosivities cannot be split between days, while daily values were used for rainfall and temperature. In cases, where an event extended over more than one day, the data of the day with the largest rainfall were used. In consequence, the equation returns event erosivities. The discrepancy between event rainfall and daily rainfall contributes to scatter in the transfer function, but its overall effect should be accounted for by the equation.

[Figure]

140 **Figure A4: Transfer function to derive rain erosivity (erosion index) from climate projections: Correlation of the measured erosion index with a prediction using the measured daily amount of rain and the daily air temperature (adopted from Auerswald et al., 2019d). The axes are square root scaled. The unit N h$^{-1}$ can be converted to MJ mm ha$^{-1}$ h$^{-1}$ by multiplying by 10.**

145 In addition to the agreement with the validation data, the spatial pattern of rain erosivity and the interannual pattern of rain erosivity, when applying the transfer function to the climate projection data of the 1970s, agreed with the published map and interannual variation of rain erosivity from rain gauge measurements in Bavaria (Rogler and Schwertmann, 1981). Also, the trend in rain erosivity calculated rain from climate projections yielded a similar behavior for the period covered by measurements (1970-2022; measured data were taken from Auerswald et al., 2019b;c; Winterrath et al., 2023), although both 150 data sets were determined independently and used different data. Specifically, the measured erosivities did not use daily air temperature and daily precipitation but temporarily highly resolved rain data. For the years 1970 to 2016, these data were

obtained from rain gauge measurements, while from 2001 to 2022, rain radar data (5 min resolution) for ca. 350000 locations were used (Auerswald et al., 2019b).

More details of the climate projections and the development of the transfer function can be found in Auerswald et al. (2019c).

155 **Table A1: Euro-Cordex climate models approved by the German Weather Service were used to model future erosivity and runoff. For abbreviations see Hübener at al. (2017).**

| Number | General circulation model (GCM) | Institution | Regional climate model (RCM) | Driving ensemble member |
|---|---|---|---|---|
| 2 | CNRM-CM5 | Centre National de Recherches Meteorologiques | ALADIN53 | r1i1p1 |
| 4 | EC-EARTH | EC-EARTH Consortium | KNMI-RACMO22E | r1i1p1 |
| 6 | EC-EARTH | EC-EARTH Consortium | CCLM 4-8-17 | r12i1p1 |
| 7 | EC-EARTH | EC-EARTH Consortium | SMHI-RCA4 | r12i1p1 |
| 9 | IPSL-CM5A-MR | Institut Pierre-Simon Laplace | SMHI-RCA4 | r1i1p1 |
| 12 | HadGEM2-ES | Met Office Hadley Centre | SMHI-RCA4 | r1i1p1 |
| 13 | MPI-ESM-LR | Max Planck Institute for Meteorology | CCLM 4-8-17 | r1i1p1 |
| 14 | MPI-ESM-LR | Max Planck Institute for Meteorology | REMO2009 | r1i1p1 |
| 15 | MPI-ESM-LR | Max Planck Institute for Meteorology | SMHI-RCA4 | r1i1p1 |
| 16 | MPI-ESM-LR | Max Planck Institute for Meteorology | REMO2009 | r2i1p1 |

**References**

Ahnert, F. (Ed.): Landforms and Landform Evolution in West Germany, Catena Supplement, 15, Cremlingen, Germany, 347 p., 1989.

160 Auerswald, K., Moyle, P., Seibert, S. P., and Geist J.: HESS opinions: Socio-economic and ecological trade-offs of flood management – benefits of a transdisciplinary approach, Hydrol. Earth Syst. Sc., 23, 1035–1044, https://doi.org/10.5194/hess-23-1035-2019, 2019a.

Auerswald, K., Fischer, F.K., Winterrath, T., and Brandhuber, R.: Rain erosivity map for Germany derived from contiguous radar rain data, Hydrol. Earth Syst. Sc. 23, 1819–1832, https://doi.org/10.5194/hess-23-1819-2019, 2019b.

165 Auerswald, K., Fischer, F., Winterrath, T., Elhaus, D., Maier, H., and Brandhuber, R.: Klimabedingte Veränderung der Regenerosivität seit 1960 und Konsequenzen für Bodenabtragsschätzungen, [Climate-change-induced changes in rain erosivity and consequences of soil loss estimation (in German)], in: Bodenschutz, Ergänzbares Handbuch der Maßnahmen und Empfehlungen für Schutz, Pflege und Sanierung von Böden, Landschaft und Grundwasser (Loseblattsammlung), edited by: Bachmann, G., König, W., Utermann, J., Berlin, Erich Schmidt Verlag, 4090, 21 p., 2019c.

170     Auerswald K., Fischer F., Winterrath T.: R-Faktor – Regenerosivität, in: Pilotstudie „Klimawirkungskarten Bayern". UmweltSpezial, Bayerisches Landesamt für Umwelt, Augsburg, 61-69, 2019d.

Baumeister, C., Gudera, T., Hergesell, M., Kampf, J., Kopp, B., Neumann, J., Schwebler, W., and Wingering M.: Entwicklung von Bodenwasserhaushalt und Grundwasserneubildung in Baden-Württemberg, Bayern, Rheinland-Pfalz und Hessen (1951-2015), [Development of soil hydrology and groundwater recharge Baden-Wuerttemberg, Bavaria, Rhineland-Palatinate and

175     Hesse (in German)], KLIWA-Berichte 21, 102 pp., https://www.kliwa.de/_download/KLIWAHeft21.pdf, last access: 08 March 2024, 2017.

BStMWi: Bayerns Wirtschaft in Zahlen 2023. Bayerisches Staatsministerium für Wirtschaft, Landesentwicklung und Energie, https://www.stmwi.bayern.de/publikationen/detail/pub-bayerns-wirtschaft-in-zahlen-2023/, last access 08 March 2024, 2023.

Bayerisches Landesamt für Statistik: Bodennutzung der landwirtschaftlichen Betriebe in Bayern 2022.

180     https://www.statistik.bayern.de/mam/produkte/veroffentlichungen/statistische_berichte/c1102c_202200.pdf; last access 07 March 2024, 2023

Fischer, F.K., Winterrath, T., Auerswald K.: Temporal- and spatial-scale and positional effects on rain erosivity derived from point-scale and contiguous rain data, Hydrol. Earth Syst. Sc., 22, 6505–6518, https://doi.org/10.5194/hess-22-6505-2018, 2018.

185     Hübener, H., Bülow, K., Fooken, C., Früh, B., Hoffmann, P., Höpp, S., Keuler, K., Menz, C., Mohr, V., Radtke, K., Ramthun, H., Spekat, A., Steger, C., Toussaint, F., Warrach-Sagi, K., and Woldt, M.: ReKliEs-De Regionale Klimaprojektionen Ensemble für Deutschland, [ReKliEs-De Regional climate projections ensemble for Germany (in German)], https://doi.org/10.2312/WDCC/ReKliEsDe_Ergebnisbericht, 2017.

Jones C., Giorgi F., Asrar G.: The coordinated regional downscaling experiment: CORDEX, an international downscaling link

190     to CMIP5, Clivar Exchanges, 16, 34-40, 2011.

Klämt, A.: Langzeitverhalten von Sonnenscheindauer und Globalstrahlung sowie von Verdunstung und Klimatischer Wasserbilanz in Baden-Württemberg und Bayern, KLIWA-Berichte 12, 147 pp., https://www.kliwa.de/_download/KLIWAHeft12.pdf, last access: 08 March 2024, 2008.

Knist, S., Goergen, K., Buonomo, E., Christensen, O. B., Colette, A., Cardoso, R. M., Fealy, R., Fernández, J., García-Díez,

195     M., Jacob, D., Kartsios, S., Katragkou, E., Keuler, K., Mayer, S., van Meijgaard, E., Nikulin, G., Soares, P. M. M., Sobolowski, S., Szepszo, G., Teichmann, C., Vautard, R., Warrach-Sagi, K., Wulfmeyer, V., and Simmer, C.: Land-atmosphere coupling in EURO-CORDEX evaluation experiments, J. Geophys. Res.-Atmos., 122, 79-103, https://doi.org/10.1002/2016JD025476, 2017.

Lenderink, G. and van Meijgaard, E.: Increase in hourly precipitation extremes beyond expectations from temperature changes,

200     Nat. Geosci., 1, 511–514, https://doi.org/10.1038/ngeo262, 2008.

NRCS: Estimation of direct runoff from storm rainfall, in: National Engineering Handbook. Part 630 Hydrology, chapter 10, Natural Resources Conservation Service (NRCS), United States Department of Agriculture, pp. 79, 2004.

Winterrath, T.: Jährlicher (2001-2019) R-Faktor [N/h/yr] auf Basis der stündlichen Niederschlagszeitreihen der RADKLIM-Version 2017.002, [R factor [N/h/yr] based on hourly rainfall series of RADKLIM, version 2017.002 (in German)],

205 https://opendata.dwd.de/climate_environment/CDC/grids_germany/annual/erosivity/precip_radklim/2017_002/, last access: 15 May 2023, 2023.

Woodward, D.E., Hawkins, R.H., and Quan, Q.D.: Curve number method: Origins, applications and limitations, in: Hydrologic Modeling for the 21st Century: 2nd Federal Interagency Hydrologic Modeling Conf., Las Vegas, NV, http://ftp.bossintl.com/download/Runoff-Curve-Number-Method-Origins-Applications-and-Limitations.doc, 2002.

210 Rogler, H., Schwertmann, U.: Erosivität der Niederschläge und Isoerodentkarte Bayerns, [Erosivity of precipitation and iso-erosivity map for Bavaria], Z. Kulturtech. Flurberein. 22, 99–112, 1981.

Seibert, S. and Auerswald, K.: Hochwasserminderung im ländlichen Raum – Ein Handbuch zur quantitativen Planung, [Flood mitigation in rural areas – a handbook for quantitative planning (in German)], Springer Verlag, https://doi.org/10.1007/978-3-662-61033-6, 2020.

215 .

---

## Author Comment (AC2)

Reply to **RC3**, Adriaan J. (Ryan) Teuling, 26 Aug 2024

Dear Editor, dear Reviewer,

We would like to thank you for your time and input on our manuscript which we have now addressed in the below point-by-point responses. Please note that we did not agree with some of the suggestions as explained in detail under the respective comments. We are still grateful for the opportunity of having this discussion, which shows that our opinion paper introduces a new perspective and, exactly as hoped for, is already having the desired effect of inspiring a scientific discussion about the topic.

Our reply is printed in blue, while the comments by the reviewer are given in black.

Sincerely,

Karl Auerswald, Juergen Geist, John N. Quinton, Peter Fiener

This contribution aims to argue that factors such as land use and soil management are more important than increasing temperatures when it comes to their impact on floods and droughts. This is an interesting and perhaps controversial viewpoint that can help to stimulate the debate on the origins of hydrological extremes.

We appreciate this comment.

While I can appreciate what the authors are trying to achieve, and I second many of the points made, the contribution currently contains numerous strong claims that are insufficiently rooted in either scientific evidence or logic. The contribution also suffers from a lack of clear definitions of what a flood or drought is according to the authors, and at what scale processes are relevant. As a result, their argumentation suffers. As an example: if flooding is defined as the length of (over)saturation of soils at a given location (which is perhaps not totally unreasonable), than artificial drainage will locally lead to a reduction of flooding, and not by definition to an increase downstream. This is in fact the main reason that much of Europe has seen the introduction of intense drainage. The contribution has similar weak argumentation in other places, which in my view leads to a situation in which the main proposition (Land use, soil management, and landscape hydrology are more significant drivers than increasing temperatures) is not sufficiently backed up by evidence or strong arguments. Therefor I believe the contribution should either be toned down (Focusing on the question are Land use, soil management, and landscape hydrology are more significant drivers than increasing temperatures?), or stronger arguments should be provided to back up the claims.

We are surprised about the misunderstanding of the term' flood' by the reviewer. Flood is not an oversaturation of soil but, according to Merriam-Webster, an inundation or overflowing of normally dry land. Please note that the first sentence of our introduction reads "*Reports of severe storms, catastrophic floods like the Simbach event (Brandhuber et al., 2017; Mayr et al., 2020), and tragic events such as the Ahrtal floods, which caused over 150 casualties (Mohr et al., 2023), are increasingly common.*" From this introduction, it should be clear to the reader that we are not equating flooding with high soil moisture.

To prevent any potential misunderstanding among different hydrological disciplines, we now clarify that 'flood' in our manuscript covers fluvial foods and flashfloods. We do not define these terms, but we expect that they will be well-known to the readership of HESS. In the introduction, we now write (insertions in bold):

"*In this paper we demonstrate and compare the $CO_2$-driven and land-use-driven climate change on floods and droughts.* **Floods encompass flashfloods and fluvial floods.** *We will…*"

Detailed comments:

Line 24 and onward: Here, the authors attack the use of common statements by arguing they are solely based on correlations and not causality. This attack fails to convince. It is general knowledge that the evaporation process (the phase transitioning from liquid to gaseous phase) depends on temperature and vapor pressure deficit. This is not an example of circular reasoning. Atmospheric and soil (moisture) states are both part of a complex (and open) feedback system, and many statements on direction of feedbacks implicitly assume timescales associated with these processes. The argument on recycling, which I find weak, is a good example of mixing of scales by the authors. Soils can be dry because it hasn't rained locally, but indeed this drying might contribute to rainfall thousands of kms away. Not sure what the authors exactly try to prove here.

We are a bit surprised by this comment because the reviewer himself wrote: "*as … land evaporation is reduced, hence the air becomes even drier, which may further decrease the likelihood of rainfall and favor the occurrence of meteorological droughts*" (Miralles et al., 2019). The only difference is that Miralles et al. (2019) focus on the meteorological origins of drought while we expand the view and point to the fact that land use can also lead to restricted land evaporation, e.g., by creating impermeable surfaces.

Line 70 and onward: Here the authors argue that because summer rainfall does not show a clear trend, rainfall cannot explain any increases in floods. This is a false comparison. A fair comparison would be to analyse event precipitation that induced flooding. It is well possible, and in line with the arguments provided on erosivity, that extreme precipitation is showing a clear increase whereas summer rainfall in itself does not. Please provide a fair comparison.

Please note that we used the sum of summer rainfall (June, July, August) to show that a change in the sum of precipitation in this period does not explain increasing drought. In the supplement, we compiled the results of a long-term (1951 to 2015) hydrological modeling set up by a working group of governmental hydrological authorities and the German Weather Service ("KLIWA"; www.kliwa.de) to detect climate change influences on hydrology. This modeling, based on daily meteorological data, shows no trend in direct runoff (and evapotranspiration). We have now moved the results of this modeling from the supplement, describing the natural boundary conditions in Bavaria, to the main part of the manuscript (now Fig. 1). This also avoids the frequent reference to a figure in the supplement that we had in our original submission.

[Figure]

*Figure 1: Change in air temperature ($T_A$), falling precipitation (P), actual evapotranspiration ($ET_{act}$), direct runoff ($R_D$), and groundwater recharge ($R_G$) during the past seven decades and averaged over entire Bavaria; the data were compiled from Baumeister et al. (2017) who had applied a hydrological model (GWN-BW) particularly designed to calculate groundwater recharge. Like almost all hydrological models, the model considers only coarse land-use classes and not accounting for the changes that occurred within the land uses. Furthermore, it does not consider lateral interactions between land uses (e.g., oasis and clothesline effects on evapotranspiration, run-on infiltration, and interflow). Hence, the model results almost entirely depict meteorological changes in space and time.*

Furthermore, we now include a reference to the latest analysis of return periods of heavy rain by the German Weather Service covering 1965 to 2020, which has just appeared. This analysis applied sophisticated statistical tools to detect a $CO_2$-driven climate change signal (Willems, et al., 2023; Shehu, et al., 2023). They analyzed rainfall depth–duration–frequency curves for 12 rain durations comprising 5, 10, 15, and 30 minutes, 1, 2, 6, and 12 hours, 1, 1.5, 3, and 7 days for the years 1965 to 2020. No trend was found for all time durations and stationarity had to be assumed (Haberlandt, et al., 2023). Thus, we conclude the section about meteorological changes by stating, "These minor changes in annual rainfall, seasonal rainfall, or event rainfall do not align with the severity of floods and droughts experienced".

In contrast to the $CO_2$-driven climate change signal on precipitation, which is still not significant, the land-use-driven changes like sealing or drainage on hydrology are statistically without doubt. The only clear and pronounced $CO_2$-driven change, except for the rising temperatures, is the increase in rain erosivity, which was identified by some of the authors of this manuscript (Fiener et al., 2013; Auerswald et al., 2019). Rain erosivity, however, influences sediment detachment and transport. It may lead to surface puddling and crusting enhancing runoff, but this requires a bare soil surface and, therefore, relates to land use.

Auerswald, K., Fischer, F. K., Winterrath, T., and Brandhuber, R.: Rain erosivity map for Germany derived from contiguous radar rain data, Hydrol. Earth Syst. Sc., 23, 1819–1832, https://doi.org/10.5194/hess-23-1819-2019, 2019.

Fiener, P., Neuhaus, P., and Botschek, J.: Long-term trends in rainfall erosivity – analysis of high resolution precipitation time series (1937–2007) from Western Germany, Agr. Forest Meteorol., 171–172, 115–123, 2013.

Haberlandt, U., Shehu, B., Thiele, L., Willems, W., Stockel, H., Deutschländer, T., Junghänel, T., and Ostermöller, J.: Methodische Untersuchungen für eine Neufassung der regionalisierten Starkregenstatistik KOSTRA-DWD [Methodological investigations for updating the regionalised extreme rainfall statistics KOSTRA-DWD], Hydrologie & Wasserbewirtschaftung 67, 138-159, 2023, https://doi.org/10.5675/HyWa_2023.3_1, 2023.

Shehu, B., Willems W., Stockel H., Thiele L.B., and Haberlandt U.: Regionalisation of rainfall depth–duration–frequency curves with different data types in Germany, Hydrol. Earth Syst. Sci., 27, 1109-1132, https://doi.org/10.5194/hess-27-1109-2023, 2023.

Willems, W., Stockel, H., Haberlandt, U., Shehu, B., Junghänel, T., Ostermöller, J., and Deutschländer T.: Betrachtungen zur Instationarität extremer Niederschläge in Deutschland [Reflections on the instationarity of extreme rainfalls in Germany], Hydrologie & Wasserbewirtschaftung, 67, 151-159, 2023.

The authors use the study of Davin et al. (2014) to argue that land management would have contributed to a shorter, less intense drought. There are several problems with this paragraph. First, the authors claim that the change of albedo would have lowed temperature country-wide by 2 K. This claim is not supported by the work of Davin et al. Yes, during some individual days during the 2003 heatwave, their simulations showed a reduction in temperature maxima of close to 2K, but only averaged over pixels with more than 60% cropland (their Fig 4). These are likely not the pixels where the high death toll mentioned by the authors took place, and the effect of cropland management on these would likely have been very small. The authors also argue that this 2K difference during the warmest days would have lead to a reduction in soil drought – seemingly in contradiction with a later statement that 2K warming does not significantly affect evaporation. The authors also seem to have done some selective shopping for arguments: in Davin et al. it is also shown that for much of France (NE, see their Fig 3) the warming impact of NOTILL for lower temperature quartiles is nearly the same as the cooling impact for higher quartiles. Please study this reference (and other references) again to make sure statements are in line with evidence provided in these works.

We had written:

"*For France, it was estimated that, during the centennial European heat wave in August 2003, if the farmers had left the straw from grain harvest on the soil rather than tilling it in, the change of albedo would have lowered temperature country-wide by 2 K (Davin et al. 2014).*"

Davin et al. (2014) had written:

"*During the peak of the heat wave in August, the daily maximum temperature was 9.9 °C above the 1986–2009 climatology according to a gridded observational dataset for temperature (20) (taking an average over 10 d between the fifth and 14th of August). This figure is well reproduced by the model, with an anomaly of 10.2 °C in the CTL simulation. In simulation NOTILL, the anomaly is only of 8.4 °C owing to the effect of no-till management, which represents a 2 °C reduction of the heatwave anomaly over this 10-d period. Fig. 4 also*

*shows that the albedo increase is the dominant factor, the evaporation effect having a relatively minor role during this specific event."*

The intention of Fig. 4 in Davin et al. (2014) is not to support this 2 °C statement, but to differentiate between the effects of albedo and soil evaporation exerted by a straw cover. Davin et al. (2014) had indeed restricted their data set for this particular task because the evaporation effect is very small, but not as much as the reviewer suggests. The caption of Fig. 4 says," ... *averaged over France (44 - 50° N; -5 - 5°E), considering only grid cells with more than 60% of cropland*." The figure shows, that the albedo effect is larger (!) than the total effect because of a small antagonistic effect of soil evaporation.

Also, the argument that there are regional differences is trivial. Yes, there are regions with a lower temperature decrease (especially along the north coast) and others that have a higher effect (especially in the northwest) to yield an average of 2 °C.

We do not see a significant discrepancy between our sentence and the detailed statement in Davin et al. (2014), but we add "on average" to our sentence to remind the reader that France is not homogeneous.

Neither the reviewer nor we know where the death toll occurred, and we did not write about this. It is evident, however, that settlements were also within the study area of Davin et al. (2014).

Line 117 and onwards: Here the authors argue that "As the droughted area heats up more than neighboring well-watered areas, the higher temperature spreads to these nearby areas and causes them to transpire more until they also run short of water. Thus, the area with reduced evapotranspiration grows and may finally spread over an entire continent, described as "event self-propagation" by Miralles et al. (2019)." Several things are wrong here. First of all, this self-propagation is never the only process operating. Please also elaborate on the role of atmospheric dynamics. And this statement seems to conflict with the arguments on recycling provided earlier. The more evaporation and drying of soils in one place, means more precipitation in other places. Again please be clear in the argumentation about spatial of temporal scales at which processes occur.

We describe what is shown in Fig. 1 by Miralles et al. (2019). It does not include atmospheric dynamics but describes, as stated by the caption, "Land feedbacks as local intensifiers of hydro-meteorological extremes," which is relevant to our topic. Of course, we agree that persistent large-scale circulation anomalies are critical for the initiation of drought and heatwaves (Miralles et al., 2019). However, how long "persistent" must be to initiate such events will depend on the moisture storage on the ground and whether it is sufficient to bridge the anomaly or not.

For clarification, we added to the text: "*Even though persistent large-scale circulation anomalies are critical for the initiation of drought and heatwaves (Miralles et al., 2019), the length of such anomalies required to initiate droughts or heatwaves will depend on the moisture storage in the ground and whether it is sufficient to bridge the anomaly or not.*"

Line 130 and onwards: Here, the authors claim based on the work by Roderick et al. (2014) that a 2 K temperature rise would increase evapotranspiration by only 5 %. This is misleading. In my understanding of the Roderick study, this is based on long-term global average values including feedbacks with precipitation. For a smaller region like Bavaria and the focus on individual droughts, the sensitivity is likely much higher. I did a quick analysis based on data for De Bilt, where potential ET (by the Makkink method) has been measured since 1957. Comparing August values for 1957-1966 and 2014-2023 results in 82.9 mm and 106.6 mm, respectively, or an increase of over 28%. The same periods show a warming of 2.67 K, resulting in a sensitivity of nearly 11 %/K, much much higher than the 2.5 %/K mentioned by the authors. This analysis took me only 5 min. I expect similar results will be found for Bavaria. Please investigate a bit further than simply picking a global number not relevant to regional drought from a 10-year old study.

Please note that we did not only cite Roderick et al. (2014) but Bürger et al. (2014) and Skliris et al. (2016) as well. This value reflects a consensus, and it does not include any precipitation feedback on evaporation. It may be that the reviewer has the impression that there would be a feedback with precipitation, because on a global scale, precipitation must equal evaporation.

The smaller than CC rate is a result of the fact that evaporation requires energy; and net irradiance does not change due to the greenhouse effect (at least as long as the effect of clouds are not well considered in climate projections). We have added a short description to the supplement, why an increase of about 2.5 %/K results from the Penman model. The FAO Penman-Monteith method is long established and is viewed as the gold standard in evaporation modelling.

The reviewer wrote "*Please investigate a bit further than simply picking a global number not relevant to regional drought from a 10-year old study.*" It is a bit surprising that the reviewer thinks that the study by Roderick et al. (2014) is too old, but then suggests to use the Makkink (1957) method. Roderick et al. was cited 199 times since 2014.  Furthermore, Roderick et al. (2014) do not focus on the global scale but they identify the drivers of local-scale changes. Remarkably, they conclude "*Much public and scientific perception about changes in the water cycle has been based on the notion that temperature enhances evaporation. That notion is partly true but has proved an unfortunate starting point because it has led to misleading conclusions about the impacts of climate change on the water cycle.*"

We tried to find a publication of De Bilt on evaporation or evapotranspiration, but Scopus returned no result when using these keywords. Please note that in the supplement, we also provide the results from an official hydrological modeling study for the years 1951 to 2015 (about the same time range as De Bilt) covering entire Bavaria. It does not show any increase in evaporation despite a 1.5 °C increase in temperature because an increase in evapotranspiration of about 4% is too small to become visible. We have moved this figure from the supplement to the main part (Fig. 1 now; see comment above) because pointing to the supplement seems to attract insufficient attention.

Line 228: "In consequence, the remaining precipitation …" -> What is remaining precipitation? Relative to what? In natural systems there will also be a considerable amount of drainage, perhaps even equal to that of drained lands (where storage on average is lower, but this does not necessarily have a strong impact on average partitioning).

We do not give a relative number.

*The full sentence reads: "According to Tetzlaff et al. (2010), 23% of the agricultural land in Germany has been artificially drained, and drainage runoff in the southern part of Bavaria can be up to 500 mm yr$^{-1}$ (Wolters et al., 2023). In consequence, the remaining precipitation on the drained land in the landscapes with the highest rainfall is as low as the precipitation in the landscapes with the lowest rainfall."*

What we are saying is that in a landscape with 1100 mm yr$^{-1}$ precipitation, where 500 mm yr$^{-1}$ is lost via artificial drainage, the difference is 600 mm yr$^{-1}$. This is the remaining amount of water available for all other hydrological processes, and this amount is equal to the total precipitation in the driest parts of Bavaria.

We rephrased the sentence to make it clearer. It now reads:

*"In consequence, after subtracting the water lost by artificial drainage the remaining precipitation in the south, which has the highest rainfall in Bavaria, is as low as the total precipitation in northern Bavaria where the rainfall is lowest."*

Section 3.4: This is a good example of selective argumentation. When discussing the study of Davin, the authors argued that a small reduction in warming would help to counteract drought. Using the same reasoning, hedgerows, which as the authors correctly point out have a warming impact on their environment (besides many other advantages!) should lead to an increase in drought because of the higher temperatures. I personally don't think this effect is very strong (and if so it is likely beneficial), but for this contribution it is important that arguments are consistent.

No; the reviewer is not correct. Based on Davin et al. (2014) we argued that a straw cover can mitigate a heatwave.

The increase in temperature due to hedgerows is an effect of reduced evapotranspiration due to less wind (allocating more energy to sensible heat at the expense of latent heat). This is simply a result of eqn 2. It cannot be argued that the increase in temperature will increase evaporation because a hedgerow does not provide any additional energy that could increase sensible heat and latent heat at the same time.

---

## Author Comment (AC3)

**RC1**: ['Comment on egusphere-2024-1702'](), Anonymous Referee #1, 08 Jul 2024 [reply]()

Auerswald et al. HESS

General comments:

The authors present a provocative piece that positions land and soil use processes at the landscape scale as important drivers of the climate change effects on floods and droughts. The manuscript reviews the relevant processes and makes a convincing argument. Only at the beginning is the piece set up in a way that suggests landscape factors to be more important than CO2-driven climate change effects. I'm not convinced by this framing, especially as the relevant evapotranspiration argument is not fully explained. I suggest positioning the two strands of impacts – via CO2 and via landscape factors – as complementary, and maybe the landscape processes have been overlooked. But the manuscript doesn't disentangle the two in my opinion. It would also be good to triangulate the arguments with timeseries data, specifically ET data from eddy-flux towers or lysimeters, soil sealing timeseries, soil compaction timeseries. I know especially the latter two are hard to come by, but maybe for the case study in Bavaria.

Unfortunately, we did not fully convince the reviewer that, at present, the $CO_2$-driven climate change is much weaker than the land-use-driven climate change because he had not found the long-term hydrological modeling in the supplement (see his second review submission). This hydrological modeling was set up by a working group of governmental hydrological authorities and the German Weather Service ("KLIWA"; www.kliwa.de) to detect climate change influences on hydrology. Now, we moved the results of this modeling from the supplement, describing the natural boundary conditions in Bavaria, to the main part of the manuscript. This also avoids the frequent reference to a figure in the supplement that we had in our original submission.

Furthermore, we now include a reference to the latest analysis of return periods of heavy rain by the German Weather Service covering 1960 to 2020, which has just appeared. This analysis applied sophisticated statistical tools to detect a $CO_2$-driven climate change signal (Willems, et al., 2023; Shehu, et al., 2023). However, the data until 2020 did not clearly show such a signal, and stationarity still had to be assumed (Haberlandt, et al., 2023). Thus, we conclude the section about meteorological changes by stating, "These minor changes in annual rainfall, seasonal rainfall, or event rainfall do not align with the severity of floods and droughts experienced".

In contrast to the still lacking $CO_2$-driven climate change signal on hydrology, the land-use-driven changes like sealing or drainage are statistically without doubt. The only clear and pronounced $CO_2$-driven change except the rising temperatures is the increase in rain erosivity, which was identified by some of the authors of this manuscript (Fiener et al., 2013, Auerswald et al., 2019). Rain erosivity, however, quantifies sediment detachment and transport.

We do not include data on eddy covariance measurement for two reasons. (i) Long-term measurements that could detect a climate change signal do not exist. (ii) A climate change signal would say nothing about whether it is derived from $CO_2$ effects or land-use effects. In fact, eddy covariance measurements are strongly influenced by land-use effects due to the

large fetch. We have therefore included references to eddy covariance results in the chapter where we describe the effects of horizontal energy advect in heterogeneous landscapes.

The request for eddy covariance data points to the central message of this manuscript. Changes in hydrological behavior, such as those potentially identified by eddy covariance measurement, cannot prove $CO_2$-driven effects without meticulously considering all other effects as well. Premature assignments to $CO_2$ effects without considering other effects can be frequently found in hydrology. Still, it is not enough to show that $CO_2$ increases over time and that some hydrological parameters also change over time. We need clear and quantitative cause-and-effect analyses in order not to overlook important drivers.

Auerswald, K., Fischer, F. K., Winterrath, T., and Brandhuber, R.: Rain erosivity map for Germany derived from contiguous radar rain data, Hydrol. Earth Syst. Sc., 23, 1819–1832, https://doi.org/10.5194/hess-23-1819-2019, 2019.

Fiener, P., Neuhaus, P., and Botschek, J.: Long-term trends in rainfall erosivity – analysis of high resolution precipitation time series (1937–2007) from Western Germany, Agr. Forest Meteorol., 171–172, 115–123, 2013.

Haberlandt, U., Shehu, B., Thiele, L., Willems, W., Stockel, H., Deutschländer, T., Junghänel, T., and Ostermöller, J.: Methodische Untersuchungen für eine Neufassung der regionalisierten Starkregenstatistik KOSTRA-DWD [Methodological investigations for updating the regionalised extreme rainfall statistics KOSTRA-DWD], Hydrologie & Wasserbewirtschaftung 67, 138-159, 2023, https://doi.org/10.5675/HyWa_2023.3_1, 2023.

Shehu, B., Willems W., Stockel H., Thiele L.B., and Haberlandt U.; Regionalisation of rainfall depth–duration–frequency curves with different data types in Germany, Hydrol. Earth Syst. Sci., 27, 1109-1132, 2023.

Willems, W., Stockel, H., Haberlandt, U., Shehu, B., Junghänel, T., Ostermöller, J., and Deutschländer T.: Betrachtungen zur Instationarität extremer Niederschläge in Deutschland [Reflections on the instationarity of extreme rainfalls in Germany], Hydrologie & Wasserbewirtschaftung, 67, 151-159, 2023.

Specific comments:

L23: The "imbalance" suggested here needs more explanation.

We replaced imbalance by alteration.

L49-61: In this section, land use and soil use seem to be conflated. I suggest to be clear about the focus.

The paragraph, like the entire manuscript, covers both: changes in land use, like converting cropland to urban land, and in soil use, like tilling straw in or leaving it on the surface. We think that our wording is correct.

L127-129: The argument for a small effect of rising T on ET is crucial and hence needs more explanation. Maybe unpack the Penman-Monteith equation to pinpoint all the influences of T

on ET and then convincingly show how these may be smaller than conventional wisdom holds.

We follow the reviewer's suggestion and include a chapter in the supplement on the Penman equation of evapotranspiration to explain why an increase in vapor pressure deficit caused by a temperature increase roughly results in a less than half as strong increase in evapotranspiration. This is, however, a very simplistic, stationary view that neglects feedback mechanisms on a global scale (e.g., decreasing wind with increasing polar temperatures). This small effect is unequivocal in the climate-change community and is the outcome of all climate projections. We had already cited three publications but added another.

Lambert, F.H., and Webb, M.J.: Dependency of global mean precipitation on surface temperature, Geophys. Res. Letters, 35, L16706, doi:10.1029/2008GL034838, 2008.

Figure 3 also misses the ET argument.

The reviewer is correct. Integrating ET into the graph is difficult because other arrows would be required. Hence, we changed the caption of the figure.

If we draw an analogous graph for ET, the effect of land use would be twofold (increasing temperature and decreasing air humidity due to unvegetated surfaces or drained, formerly wet spots). In contrast, the CO2 effect only increases the temperature.

L142-143: How to disentangle CO2-driven climate change from land use-driven climate change?

This is difficult, and it is work in this is what we want to stimulate. As long as we are not able to disentangle both drivers, it is improper to assign effects to one of both causes. This is, however, frequently done with potentially misleading results.

In some cases, disentangling is relatively easy. For example, the influence of land use on heavy rain should be minimal (but for this parameter, we still do not see significant effects; see above), while land use and $CO_2$-driven climate change will influence runoff and flooding. In the public discussion, flooding is presently almost exclusively associated with CO-driven climate change despite the lack of measurable effects on heavy rain. More efforts of science to disentangle both drivers are urgently needed.

We did not make a change here because the question raised by the reviewer will require many publications, and our manuscript also raises it.

L150-151: What about open water evaporation from ponding on sealed surfaces?

We modified the text. It reads now: "Except for the small amount of water left on sealed surfaces after a rain, sealed surfaces do not contribute to evaporation but partition their radiant energy uptake almost exclusively into sensible heat."

Some of the reference in section 3.1 are also quite old. Do recent studies confirm these findings?

We deleted Oke (1989), which we had put in to honor the pioneering work of Oke, but we keep Oke (1982), Calder (1949), Drivas and Shair (1974), and McNaughton (1976, 1978). There are no newer studies that would invalidate these old studies and justify removing them. More importantly, these old studies show that we have known the effects of energy advection in great detail for many decades but do mostly not include them in hydrological modeling or consider them in landscape planning. We had also cited newer references from 1995, 1996, 2000, 2007, 2018, 2021. We now expanded this chapter to include more recent publications. In particular, we also added references to the vast literature on eddy covariance measurement, for which the lateral effects of advection are central (called fetch in this context). And, we added the motivation to use old references.

Baldocchi, D.D. and Rao, K.S.: Intra-field variability of scalar flux densities across a transition between a desert and an irrigated potato field, Boundary-Layer Meteorol., 76, 109–136, 1995.

Grossiord, C., Buckley, T.N., Cernusak, L.A., Novick, K.A., Poulter, B., Siegwolf, R.T.W., Sperry, J.S. and McDowell, N.G.: Plant responses to rising vapor pressure deficit, New Phytol, 226, 1550-1566, https://doi.org/10.1111/nph.16485, 2020.

Klaassen, W., Van Breugel, P.B., Moors, E.J. and Nieveen, J.P.: Increased heat fluxes near a forest edge, Theor. Appl. Climatol, 72, 231–243, 2002.

Leclerc, M.Y. and Thurtell, G.W.: Footprint prediction of scalar fluxes using a Markovian analysis, Boundary-Layer Meteorol., 52, 247–258, 1990.

Panin, G., Tetzlaff, G. and Raabe, A.: Inhomogeneity of the land surface and problems in the parameterization of surface fluxes in natural conditions, Theor. Appl. Climatol., 60, 163–178, https://doi.org/10.1007/s007040050041, 1998.

Raatz, L., Bacchi, N., Pirhofer Walzl, K., Glemnitz M., Müller M.E.H., Joshi, J., and Scherber, C.: How much do we really lose?—Yield losses in the proximity of natural landscape elements in agricultural landscapes, Ecology Evolution, 9, 7838–7848, https://doi.org/10.1002/ece3.5370, 2019.

Savage, M.J., McInnes, K.J., and Heilman, J.L.: The 'footprints' of eddy correlation sensible heat flux density, and other micrometeorological measurements. South African J. Sci., 92, 137–142, 1996.

And how do the timelines of $CO_2$ in the atmosphere and soil sealing compare? Does a comparison support the argument of a soil sealing-driven climate change?

This comment has three independent aspects. (i) Both, $CO_2$ and soil sealing continuously increase. Hence they correlate closely as many other timelines do. (ii) In this specific case, there is not only a correlation but also a causal relation because more sealing means more traffic and more cement production and, in turn, more $CO_2$ release. (iii) In this manuscript we do not focus on the $CO_2$ release by sealing but on the climate effect of sealing caused by the modified allocation of radiation energy to heat and evaporation.

No change was made to the manuscript.

L244-248: What about outflow from the river to the groundwater?

Usually and initially, we have gaining rivers in Bavaria that should receive their water from the groundwater. However, due to the widespread lowering of the groundwater, loosing rivers were created. Consequently, many small rivers have disappeared while others have become perched due to colmation (clogging) of the river bed.

We added "…and caused many small rivers to disappear (Reckendörfer et al., 2013, Zerbe, 2013)"

L264-266: How does soil compaction look over time?

This is an open question because no consistent monitoring over decades exists. However, in a recent German-wide survey in an 8×8 km grid analyzing 16778 soil samples, 51% of the arable land had root restrictions caused by compaction (Schneider and Don, 2019). This value is similar to the 43% of overcompacted soils found in the Netherlands (Brus and van den Akker, 2018).

We now include reference to the work of Schneider and Don (2019) and Brus and van den Akker (2018) in the revised version of the manuscript.

Brus, D. J. and van den Akker J. J. H.: How serious a problem is subsoil compaction in the Netherlands? A survey based on probability sampling, Soil, 4, 37–45, https://doi.org/10.5194/soil-4-37-2018, 2018.

Schneider, F. and Don, A.: Root-restricting layers in German agricultural soils. Part I: extent and cause, Plant Soil, 442, 433-451 (2019), https:doi.org/10.1007/sII104-019-04185-9

L342: What is meant by "unfavourable behaviour" here?

We added "like increasing complexity, instability and exponential growth or oscillations"

**Citation**: https://doi.org/10.5194/egusphere-2024-1702-RC1

---

## Author Response (AR2)

Dear Editor,

Thank you for giving us the opportunity to revise our manuscript; we hope that you find our revisions and responses to reviewers acceptable.

We hope that our opinion goes some way to addressing a deficit within the hydrological community, namely that that the land-use community and the $CO_2$ community act separately and largely ignore each other. Both communities deliver a biased explanation for the increased occurrence and severity of floods and droughts, as long as land use and $CO_2$ are not considered together. To highlight this deficit, we found it more appropriate to express our title as a question that requires to be answered in the future. This also complies with the suggestion of referee #2 that the title should better reflect the opinion character of our manuscript. The title reads now:

HESS Opinion: Floods and droughts - Are land use, soil management, and landscape hydrology more significant drivers than increasing CO2?

We hope this manuscript emphasises the need to tackle all drivers of floods and drought together and that this is a message that HESS would like to see published as an opinion..

Thank you for your consideration.

Kind regards

Karl Auerswald, Juergen Geist, John N. Quinton, and Peter Fiener
.

Dear Editor, dear Reviewers:

Below, the reviewer's comments are given in black, while our response is printed in blue. We inserted the respective parts of our previous manuscript as comments to allow an easy comparison without the need to switch between documents.

**Reviewer 1**

The authors revised their manuscript in response to the comments and suggestions of two reviewers, one of them me. The authors clarified sentences, moved figures from a modelling study and corresponding explanations from the appendix and added a new section on scale effects. They didn't modify or tone down their arguments.

We thank the reviewer for acknowledging that the adjustments improved the clarity of our paper. However, we disagree with the reviewer that modifying or toning down our arguments would improve the paper. For instance, it is well-known how land use influences hydrology. Hence, we do not believe that toning down and writing something like "sealing may increase runoff; compaction may increase runoff…" would improve an opinion paper. The disagreement only results from interpreting a complex system response like flooding or drought, when suddenly only one explanation ($CO_2$) seems to be allowed. We argue that these land-use-related mechanisms must be fully considered when explaining extremes like the Valencia flood. Should we write "land use may be considered as well"? This would clearly be wrong. According to the journal guidelines, especially opinion papers are intended to deliver clear messages and inseminate discussion. This impact would be lost if we further tried to tone down our argumentation.

I'm sympathetic to publication since this is an opinion piece that will hopefully attract scholarly debate. I would still suggest the following revisions to improve the authors' arguments. They mainly relate to their arguments against CO2-driven causes of floods and droughts, which, as reviewer 2 noted, are rather unbalanced and suffer from mixing and matching various kinds of information (data, theory, models) from various spatial and temporal scales. The authors' arguments for a greater consideration of landscape scale drivers of floods and droughts, on the other hand, are generally balanced.

Thanks for being sympathetic to the publication despite our different opinions on some of the suggested aspects – we also hope that this paper will initiate further debate on this topic. Concerning the additional suggestion for revision, we think this is too unspecific to be answered. We took great care using official data from a well-defined region. We consider it a strength of the paper that we use multiple data sources from the published literature and that we do not exclusively rely on either data, or theory, or modelling. Furthermore, mixing measured data and models is not our responsibility because we rely on the suite of published studies on this topic. Furthermore, this situation is generally unavoidable in hydrology because several parameters like groundwater recharge, evapotranspiration or runoff are usually derived from modelling, particularly when a large region and a long period is considered.

A lot of the authors' arguments hinge on the plausibility of the modelling studies they put forward to support their arguments. These models are not scrutinized at all with respect to their model structures, parameter values and input data. And none of them include the complexity the authors argue is needed! If the models only include climate drivers, which the authors use to isolate climate effects (and ultimately suggest they are minor), how can the models ever shown to be plausible?

We fully agree with the reviewer that models, which are indispensable in hydrology, are a critical point. Models can – at best – only find things that are included in the model. We then conclude that

those parameters are important, which is circular reasoning. We added a sentence on this to the conclusions.

Next, I pinpoint specific places where the argument is weak in that sense in addition to other comments.

Specific comments:

Title: I wonder whether "temperature" should be replaced with "CO2" because the authors talk in these terms in the text (CO2 is also the section 2 heading). Especially as they explain how landscape scale factors can also increase temperature. Making this change I think will make the framing of the paper clearer.

Thanks for spotting this. We agree and have changed this accordingly. Following a suggestion of Ref. #2 we also modified the entire title and present it as a question to avoid the impression of a definite answer.

L39: What is a "common relation"?

The decreasing birthrate with the decreasing number of storks. This is a special case of spurious correlations, sometimes also called confounder correlation. Such relations can often be found when comparing time series. We replaced the word by the better known but wider term "spurious relation"

Equations 1 and 2: Following a concern of reviewer 2, I suggest that the authors include a note on scale for both equations. At which time and space scales are they expected to hold?

We use the equations only for illustration purposes and not for our own calculations. The two equations are intended to illustrate the following:

1) Both equations are coupled. This applies for all systems and scales where evapotranspiration happens
2) If one parameter changes, at least a second parameter must change to close the equation. This applies for all balances.

Allen et al. recommend these equations for time scales of days to years and fields to landscapes. The parameters and how they have to be derived will differ depending on scale. We added that, in principle, the equations apply to all scales but the relative importance of the different terms changes with temporal and spatial scale.

L50: The discussion here seems unbalanced: In addition to R_nl, CO2-driven climate change also impacts all other factors via cascading effects, whereby CO2-effects interact with landscape scale factors.

We disagree on this point. The discussion is perfectly balanced because we name only those parameters that are directly influenced, while we neglect all cascading parameters. Hence, we treat $CO_2$ and land use equally. However, we explicitly state now, that cascading effects are ignored.

L55: From the response to reviewer 2 I understand that the authors wanted to add "on average" here, which I find a good idea!

We added "on average" after country-wide

L67-69: Here, the authors jump to quickly from a "typical Mid-European setting" (if one accepts that Bavaria is typical for Mid-Europe) to "will occur globally", even if they note regional differences afterwards. There is no evidence provided to make such a statement. Regional differences can easily overwhelm any similarity.

We mention that there are large regional differences. However, the general trend that agricultural machinery weight increases, road density increases, and soil sealing increases can be found in many regions. We have added 'most regions' to globally as there might be settings where this in not the case.

L79: Rather than "constant energy", with reference to equation 2 the authors have noted an increasing energy input with climate change. The argument needs to be adjusted in that light.

We are not aware of any mechanism for how $CO_2$ can influence solar radiation.

L98-101: There is another contradiction here between the statement that any trend was minor compared to spatial variation (L98) and the statement that the changes were spatially relatively uniform (L101).

We do not think the was a contradiction because these are two different statements. The spatial variation of the parameters is large but the spatial variation of change is small. Both sentences are correct as they stand.

Figure 1: The model results here are given on a decadal scale and extend until 2015 only. First of all, the model needs greater scrutiny before the reader can place any trust in it. Second, important variations might be hidden in the decadal figures, i.e. floods and droughts. It would also be relevant to see the results for the most recent 9 years.

Please note that these are not our results. Hence we cannot change the resolution or the period that was considered. How likely is it that nine years in a 65-yr period would make a large change, in particular as these recent years did not differ much regarding precipitation as shown in the following figure. Temperature has risen further but this increase can already be nicely seen until 2015.

L113: It is not demonstrated in the paper that these years show no CO2-related pattern. The authors should either present this analysis or quote it if this was done somewhere else or delete the statement.

We can assume that every reader roughly knows that $CO_2$ increases over time with highest concentrations in recent years. We show in this sentence, that there is no trend to drier years or wetter years. The same can be seen in Figure 1 and Figure 2. Providing $CO_2$ concentrations would not change our sentence.

From the driest to the wettest of the five driest years, [$CO_2$] was: 300, 378, 311, 376, 309 ppm
From the wettest to the driest of the five wettest years [$CO_2$] was: 305, 304, 319, 312, 310 ppm

There is no descending or ascending order of [$CO_2$] when approaching the average. Furthermore, between 1900 and today, [$CO_2$] ranges from 300 to 425 ppm. The high concentrations do not fall together with the five driest or five wettest years.

L113-114: How is the winter precipitation trend related to CO2? The authors should analyse all these aspects symmetrically, at the same level of detail.

If desired, we can delete the part after "while". It has a marginal influence on our manuscript. We would prefer this over expanding. Expanding land use effects would be much more valuable because the changes that happened there are orders of magnitude larger and apparently widely unknown.

L121-122: I suspect the authors of that study did a Null hypothesis test of the trend and could not reject the Null. But this is not the same as no trend! And what exactly where these "sophisticated statistical tools"? This is another example of an unbalanced (here uncritical) discussion. For the reader to understand the argument, these studies need to be scrutinised at the same level of detail as other studies that the authors are more critical of.

These are published studies that the German Weather Service released. They provide the base for all planning in Germany that requires return periods. We were extremely critical in this case because of the relevance. The implications by the reviewer are not justified. Shehu et al (2023), appeared in HESS, it has nine (!) pages of statistical methods. Explaining them would completely distract from our topic with no additional insight. Nevertheless, we especially recommend reading Shehu et al. to all interested in rain data because they were able to identify shortcomings of rain-gauge measurements that could falsely lead to a trend in heavy rainfall. Willems et al. go even more into detail in this respect with illustrative figures but this is unfortunately in German.

Figure 2: Please specify what the black line is. The running median? And haven't the authors complemented the graph with years 2000 to 2024 (not 2020 to 2024)? And why complement at all? This makes the graph confusing. Can the calculations not be repeated with the original and extended dataset? This would make it easier to understand for the reader and help the argument.

The black line was a running 30-yr mean that we included because it was in the original publication. It has no relevance and we will delete it. Adding the years 2020 to 2024, which were later than the study, has the purpose to have a wider overlap of measured data with climate projections to show the good agreement between both. The original data and the complemented data are homogeneous because all data were taken from the same source (published country-means by the German Weather Service). We will additionally cite this source.

Please note that your critique of the previous figure was the opposite to your critique here. In Figure 1 we were not able to add years after the study appeared because of the complex modelling that had been used in this study.

L152: Here is another example that calls for a more symmetrical analysis: How was "the only parameter" determined? How was the CO2-driven climate change signal (or lack thereof) determined? With a model? With data?

The titles of the references clearly show that these are measured data. It is important to note that both publications used independent data and approaches. The influence of the $CO_2$-driven signal was (also)

analyzed in several follow-up publications that we do not cite to avoid inflating self-citations. The mechanisms relating to $CO_2$ are rather complex and irrelevant for this manuscript. One mechanism is that winter precipitation changes increasingly from snow to rain, which causes a pronounced change in the seasonality of erosivity. To make it clearer for the reader we added "The only measured parameter …"

Figure 3: For symmetry, this figure, too, would benefit from a trend analysis. From the appendix I understand that runoff was calculated with the SCS curve number method and erosivity with a transfer function from precipitation and temperature. This should be briefly mentioned in the caption and/or text, even if the appendix is referred to for details. For a balanced analysis, I would also expect a discussion of the limitations of those methods (some of this is already in the appendix for SCS). The erosivity model must be sensitive to the exact values of the exponents of the P and T effects (since small changes amplify through the power law), which is downplayed by the logscale of Figure A3.

The figure superimposes measured (historic) values (in cases of erosivity) or data from official, up-to-date modeling (in the case of direct runoff) based on measurements with modelling using data from climate projections. The behavior that we describe is fully visible in the historic data. If desired, we can delete the data derived from climate projections. This would have no effect on the entire manuscript, which focusses on the explanation of droughts and floods in the past.

The trend analyses exist. For the historic data they are published. They cannot show anything else than the very clear "raw" data.

L170-171: This statement the authors should qualify by adding "based on model simulations".

The reviewer is not correct. There is no modeling involved. The change of rain erosivity is based on measurements and the still undetectable change in the return periods of heavy rain is also exclusively based on measurements.

L187: This contradicts with statements in the previous paragraph where in the chain of events triggered by increasing erosivity runoff was argued to increase!

Yes, there is a contradiction but still, our sentence is correct. In hydrological modelling with the SCS curve number but also with other common approaches, rain erosivity is not considered. Ironically, the reviewer criticizes our conclusion on line 480 (see below) that runoff modeling should take soil crusting and infiltration-excess runoff into account. Furthermore, an increase in rain erosivity does not cause more runoff as long as the soil is protected from crusting. An increase in runoff due to increasing erosivity has to be assigned to a land use that does not keep the soil covered.

L190-191: Arguably the CO2-driven ET effect is subject to the same self-intensifying effects described in the previous paragraph. Even if the CO2-driven ET effect is modest to begin with.

In an arid environment, this may be the case, but in a humid area with functional soils, it will be extremely rare that a 5% increase in ET will deplete the soil. As long as this is not the case, the self-intensification and self-propagation is not initiated. We clarified:

"Therefore, a 2 K temperature rise would increase evapotranspiration by only 5 %, **which should be buffered by functioning soils in humid areas**. Consequently…"

L191-193: In L191 it says 2-3%, in L193 5%. Is this a typo? Or what is the difference?

No, this is correct. The reviewer probably missed that the unit of "2-3" is %/K while the unit of the second number is % because it was derived from the first number by multiplication with the change in temperature.

However, we adapted the sentence a bit to make the relation between the 2-3 % evapotranspiration per K and the increase in evapotranspiration in case of a temperature increase of 2 K clearer.

" … evapotranspiration rises only modestly by 2 to 3 % K-1 temperature increase (Lambert and Webb, 2008; Roderick et al., 2014; Bürger et al., 2014; Skliris et al., 2016). Therefore, a 2 K temperature rise would increase evapotranspiration by only **4 to 6** %...."

Figure 4: The authors modified the figure in response to my comment, but I still don't understand why the crucial ET argument is missing. At the top of page 8, the argument is put forward quite succinctly. Could this be transferred to the figure?

We had added a chapter to the supplement dealing with the ET argument, which shows, in accordance with the cited publications, why the $CO_2$-driven effect is small (2 to 3% per K). The reviewer seems to expect a much larger effect.

What we describe on top of page 8 has nothing to do with the $CO_2$-driven effect but it describes the self-propagation and self-intensification of drought in landscapes that are poorly buffered. At least in central Europe, this is caused by dysfunctional soils (due to drainage, compaction, sealing, poor cover). The increase in runoff causes the drought. This is included in the figure.

L218: The modelled figure of 528mm ETa per year must already account for some of the sealing historically, even if implicitly by adjusting other processes to fit the historical observations. How can we otherwise trust what the model predicts? The analysis seems to be too simplistic here, placing too much trust in a particular model to support an argument, without scrutinising the model.

ET modelling is usually based entirely on meteorological parameters without accounting for the reasons that shape these parameters. If dry surfaces cause temperature to increase and humidity to decrease, this will increase modelled ET. Hence the sealed surfaces are included without explicitly accounting for them. In consequence, the effects of sealed surfaces are usually overlooked and it will be tricky to disentangle the influences. We did not model ET but we clearly stated that we just made simple balance considerations to show that the effect is not negligible and deserves disentangling.

The surprising thing is that our simple balance consideration reproduces the gap groundwater recharge. The Bavarian Environmental Agency, responsible for the observation wells, find a similar decrease in groundwater recharge although rainfall has not changed.

L258-259: I believe it should be 6% in both instances. As above with ETa, the recharge figure of 206 mm per year must already include the sealing to some extent, even if implicitly through effective parameters.

Yes, you are correct – thanks for spotting this. We had replaced an older publication stating 5% sealing by a newer publication stating 6%, and we had forgotten to adjust all numbers.

Remarkably, both publications were from the same group applying similar methods. They are only 10 years apart but still, soil sealing has relatively increased by 20%. This also illustrates the incredibly fast changes in land use that requires attention.

L261: I don't understand from the text how the 44mm come about.

We have modified the text to improve clarity:

Furthermore, sealed areas impede groundwater recharge. **Six** percent sealing reduces the overall mean groundwater recharge (206 m yr-1, Baumeister et al., 2017) by 12 mm yr$^{-1}$. Neighboring areas, if they compensate for the loss **of 32 mm yr$^{-1}$** in evapotranspiration (Blumröder et al., 2021; Herbst et al., 2007), will, in consequence, recharge **32 mm yr$^{-1}$** less groundwater. Ultimately, this may lead to a calculated 44 mm yr$^{-1}$ decrease in groundwater recharge if vegetated surfaces compensate for the entire loss of evaporation caused by sealed surfaces.

L407-409: This statement misses a reference.

The statement was explained in the following sentence. To address the comment, we now reversed the sentences and now write:

Rain cells or pressure systems move over a location within hours or days, while subsoil compaction can persist for years or even centuries, tile drainage can remain functional for many decades or longer, and soil sealing is rarely reversed. Hence, on a temporal scale, these influences of land use last considerably longer than weather phenomena.

L455: The authors didn't "illustrate" any relative importance in this paper, so the statement "at least equally important" should be clearly labelled as speculation.

We changed the sentence and also revisited the new title. It reads now (changes in bold):

However, exclusively focusing on this goal would ignore other important mitigation measures that urgently need to be realized. As illustrated here, restoring hydrologically functional landscapes and soils **should be considered** at least equally important to mitigate climate change, especially concerning extremes such as floods, droughts, and heatwaves **and to preserve the foundation of food and life**. **The question of whether land use or $CO_2$ are more significant drivers of floods and drought deserves more attention even though a simple answer will never be possible.**

L480: The caution the authors advise here when it comes to models should be exercised by the authors themselves when they use models to support their argument.

We did not use models ourselves but, indeed, we cited papers that used models. Most of them have appeared in respected journals and we can hardly pinpoint all deficits that may be in the models. Our argument was clearly different: The effects caused by $CO_2$-driven climate change call for a reappraisal of modeling approaches. We see nothing wrong in this statement.

**Reviewer 2**

The authors have made considerable changes with respect to the original version. While I appreciate their efforts, I do not in all cases agree with their response, given that on several issues the authors try to argue their way out rather than improving the argumentation. However for a HESS Opinions contribution, I think it should be acceptable that a certain level of disagreement will remain between authors and reviewers. In my view this is fine, and since I believe the underlying message in the manuscript is important, I am willing to overlook some of the more minor issues.

We thank the reviewer for taking the time and effort spent with our paper. We also appreciate the openness towards allowing us to present our "opinion" supported by an evidence-based chain of arguments.

In spite of having spent considerable time reading and thinking about this work, it only appeared to me last week what in the overall pattern in, and reason for, my comments on the previous version in fact is. A HESS Opinions contribution is, in my view, a discussion on a topic with the aim to raise awareness of an issue that is typically understudied/overlooked where a case is built by the authors on arguments. This is, by its very nature, not an objective process. My main problem is already reflected in the title of the contribution: "… are more significant drivers …". This wording does not reflect an opinion, but something that would normally be concluded based on a model study or controlled field experiment. These are the only possible ways to formally test whether "land use, soil management, and landscape hydrology are more significant drivers than increasing temperatures". This can never be "proven" based on arguments. So the title, and some of the main conclusions (for instance "is is at least equally important"), are problematic since they do not reflect that opiniated nature of the contribution. Luckily, this is an issue that is rather easy to solve. Overall, I believe the authors need to change the phrasing from essentially suggesting that we know everything there is to know about land use and management and its importance relative to that of climate change, to presenting the referenced studies as rare examples that suggest that land use and land management (and lateral interaction) are being overlooked and might in fact be much more important drivers of hydrological change as currently believed. From this, the authors' main opinion can be that those aspects should not be overlooked (which I would fully support), and that more research is urgently needed to provide a formal answer to the questions of what is their relative importance (a call for agenda setting). Possible new titles that would fit with this approach could be "HESS Opinion: Land use, soil management, and landscape hydrology should not be overlooked as main drivers of changes in floods and droughts" or "HESS Opinion: The need to consider land use, soil management, and landscape hydrology as main drivers of changes in floods and droughts". This opinion would be defendable based on the evidence discussed, and it would solve some of the issues that might be seen as conflicting or where I disagree with the interpretation of the authors (such as the discussion on hedges).

We changed the title by asking a question to accentuate that we do not give an answer,  but encourage a discussion. Furthermore, "temperature" was replaced by $CO_2$ following the suggestion of Ref. #1.

HESS Opinion: Are changes in land use, soil management, and landscape hydrology more significant drivers of floods and drought than increasing $CO_2$?

In terms of textual changes needed to accommodate this suggestion, these could likely be minor. I leave it up to the authors to double-check the formulations in the main text. In the conclusions, it is

important that phrases such as "is at least equally important" are removed or changed since these are not directly and formally supported by the evidence. Ideally, the conclusions would end with a clear recommendation or call for agenda setting.

We changed the sentence and also revisited the new title. It reads now (changes in bold):

However, exclusively focusing on this goal would ignore other important mitigation measures that urgently need to be realized. As illustrated here, restoring hydrologically functional landscapes and soils **should be considered**  equally important to mitigate climate change, especially concerning extremes such as floods, droughts, and heatwaves **and to preserve the foundation of food and life**. **The question of whether land use or $CO_2$ are more significant drivers of floods and drought deserves more attention even though a simple answer will never be possible.**

We have also carefully checked the entire text again to avoid any parts that were misleading or not fully supported by our data. This also included explaining some steps in more detail (see response to referee #1).

I hope the authors see the value of these comments, and can make the changes that would, in my view, make this contribution to a strong and important HESS paper.

---

## Author Response (AR3)

Dear Editor,

Thank you for your suggestions on improving the clarity of the final paragraph of the discussion, we have reworded it to read as follows:

In this opinion paper, we compare the effects of $CO_2$-driven and land-use-driven climate change on floods and droughts and highlight the significant, but rarely considered, effects of land management on landscape hydrology. Floods encompass flashfloods and fluvial floods. We will exemplify this for Bavaria (southern Germany) for two reasons. (i) We have access to a large consistent monitoring and modelling data set regarding climate, land use and management and hydrology, and (ii) the region represents a typical Mid-European setting with predominantly agricultural land use (cropland and livestock farming) while also featuring extensive forest, urban areas, and protected natural reserves. We argue that effects shown here for southern Germany will occur globally, although, there will be regional differences in land use setting and agricultural management, for example, differences in agricultural machinery weights, the soil sealing area, or the road density. A short description of the example area can be found in the supplement.

Hopefully this is now clearer.  .

Kind regards

Karl Auerswald, Juergen Geist, John N. Quinton, and Peter Fiener
.